# PROSPERO: Active Learning for Robust Protein Design Beyond Wild-Type Neighborhoods

Michal Kmicikiewicz[1,2]        Vincent Fortuin[2,3,4]        Ewa Szczurek[1,5]

[1]Institute of AI for Health, Helmholtz Munich
[2]School of Computation, Information and Technology, Technical University of Munich
[3]Helmholtz AI, [4]Munich Center for Machine Learning
[5]Faculty of Mathematics, Informatics and Mechanics, University of Warsaw
{michal.kmicikiewicz, vincent.fortuin, ewa.szczurek}@helmholtz-munich.de

## Abstract

Designing protein sequences of both high fitness and novelty is a challenging task in data-efficient protein engineering. Exploration beyond wild-type neighborhoods often leads to biologically implausible sequences or relies on surrogate models that lose fidelity in novel regions. Here, we propose PROSPERO, an active learning framework in which a frozen pre-trained generative model is guided by a surrogate updated from oracle feedback. By integrating fitness-relevant residue selection with biologically-constrained Sequential Monte Carlo sampling, our approach enables exploration beyond wild-type neighborhoods while preserving biological plausibility. We show that our framework remains effective even when the surrogate is misspecified. PROSPERO consistently outperforms or matches existing methods across diverse protein engineering tasks, retrieving sequences of both high fitness and novelty.

## 1 Introduction

Proteins are essential macromolecules that play a central role in virtually all biological processes. The ability to design novel protein sequences with desired functional properties is crucial for a wide range of applications, including drug design, industrial biotechnology, and beyond [1–3]. Despite this promise, optimization of protein sequences remains a grand challenge in computational biology. The protein *fitness landscape* [4], mapping between the space of sequences and *fitness*, their corresponding functional levels, is typically rugged, sparse, and highly non-convex [5, 6]. Moreover, upon proposing a candidate sequence from the combinatorially large search space, evaluation requires querying an expensive black-box objective function. To alleviate this, machine learning models are often used as inexpensive surrogate models that approximate the costly black-box oracle [7–9]. To further facilitate navigation of the landscape, optimization commonly begins from a *wild-type* sequence, preserved through natural evolution and, as such, exhibiting reasonable fitness [6, 10].

Numerous strategies have emerged to traverse protein fitness landscapes. Ren et al. [6] introduced PEX, an evolutionary algorithm that exploits the wild-type neighborhood. While highly effective and favoring biologically plausible sequences thanks to its local focus, this approach limits broader exploration of the fitness landscape, potentially missing advantageous sequences that are inaccessible through local mutations. To overcome this, reinforcement learning (RL) methods [8, 10] and Generative Flow Networks (GFNs) [9] have been employed to target novel regions of the search space. However, these global exploration strategies often encounter issues with surrogate model

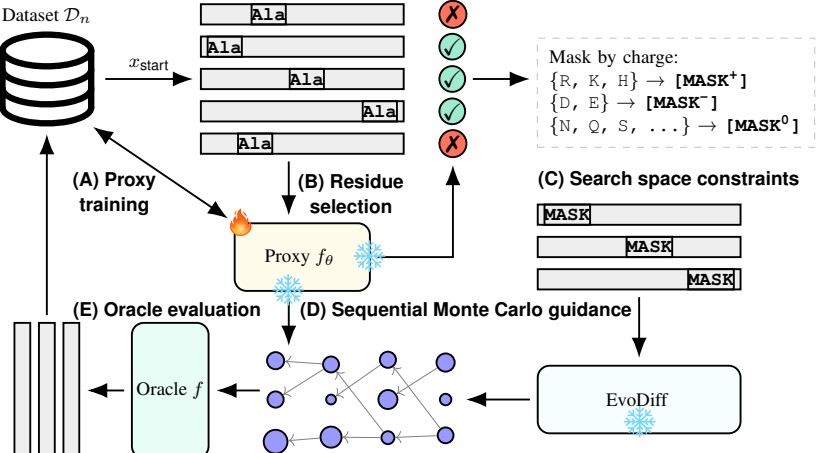

Figure 1: Overview of PROSPERO. Each active learning iteration begins with training a surrogate model on the current dataset (A). The surrogate is then used to identify fitness-relevant residues within the top sequence (B), which are subsequently masked, yielding partially masked sequences (C). EvoDiff, guided by the surrogate, completes these sequences to generate new candidates (D), which are evaluated by the oracle and added to the dataset (E).

misspecification when evaluating sequences substantially different from the surrogate's original training distribution [11]. To address this and balance exploration with robustness, GFN-AL-$\delta$CS [12] learns an unmasking policy that reconstructs partially masked sequences. Yet, since masking is applied at random, this approach may modify conserved residues, potentially degrading structural and functional integrity and yielding biologically implausible proteins. Pre-trained generative models offer a compelling alternative, inherently encoding rich biological priors that greatly reduce the risk of generating implausible sequences [13]. However, effectively incorporating such models in iterative optimization workflows presents a challenge, as each iteration would require impractical task-specific fine-tuning on limited oracle-annotated data or low-fidelity surrogate-annotated data [14, 15]. The shortcomings of existing approaches point to the need for a *protein design framework capable of generating high-fitness sequences beyond the wild-type neighborhood, while addressing the surrogate misspecification and loss of biological plausibility* that often arise from exploring such novel regions.

To meet these challenges, we introduce PROSPERO. Our main contributions are:

**(i) A robust exploration framework**, to our knowledge the first to formulate iterative design of protein sequences as inference-time guidance of a pre-trained generative model by a surrogate updated in an active learning loop. This enables straightforward incorporation of biological priors encoded by the generative model, helping preserve biological plausibility even when surrogate-guided exploration extends beyond wild-type neighborhoods.

**(ii) A targeted masking strategy**, which focuses edits on fitness-relevant residues while preserving structurally and functionally important sites. In contrast, prior approaches risk disrupting essential residues through random or uninformed masking.

**(iii) Biologically-constrained Sequential Monte Carlo (SMC) sampling**, offering a novel strategy to incorporate explicit biological priors into inference-time guidance in discrete sequence space. Restricting proposals to amino acids with properties similar to their wild-type counterparts increases the likelihood of retrieving high-fitness sequences in novel regions of the search space, where the surrogate may be misspecified.

We conduct extensive experiments across diverse protein fitness landscapes and demonstrate that PROSPERO consistently approaches the Pareto frontier between candidate sequence fitness and novelty, while preserving biological plausibility. We perform ablation studies under varying degrees of proxy misspecification to assess the contribution of individual components to the overall robustness of our method. Code is available at https://github.com/szczurek-lab/ProSpero.

## 2 Related Work

**Evolutionary Algorithms**  Evolutionary algorithms are a common approach in protein sequence design [16, 7, 6]. Notably, Sinai et al. [7] proposed a greedy algorithm, AdaLead, which adaptively mutates and recombines high-fitness sequences selected by a threshold-based filter to balance exploration and exploitation. Ren et al. [6] introduced an exploration method, PEX, designed to exploit the local neighborhood of the wild-type by prioritizing variants with fewer mutations.

**Machine-Learning-Assisted Directed Evolution (MLDE)**  To enhance traditional directed evolution [17], MLDE leverages machine learning models to predict sequence fitness and guide the selection of promising mutants for screening [1]. Qin et al. [15] iteratively fine-tune ESM-1b [18] with data annotated by a surrogate model. Tran and Hy [13] mask k-mers of a wild-type sequence and use ESM-2 [19] to propose new candidate sequences. The work of Qiu et al. [20], Qiu and Wei [21] and Wang et al. [22] explores clustering amino acids with similar properties; however, it differs from our approach by assuming a small number of fixed mutational sites.

**Reinforcement Learning and Generative Flow Networks (GFNs)**  DyNaPPO uses an ensemble of surrogate models with varying architectures to train a generative policy that constructs candidate sequences amino acid by amino acid [8]. Rather than acting in the sequence space, LatProtRL [10] learns a generative policy operating in the latent space of ESM-2 [19]. GFNs use a surrogate model to learn a stochastic policy that samples sequences proportionally to their predicted fitness values [9]; however, when the proxy is misspecified, they can perform poorly [11, 12]. To address this, Kim et al. [12] proposed GFN-AL-$\delta$CS, a strategy enabling to control a trade-off between novelty and robustness based on uncertainty of the proxy.

**Bayesian Optimization (BO)**  BO is a commonly used framework for optimizing expensive black-box functions and has been widely applied to the design of biological sequences under limited evaluation budgets [23–26]. Among these approaches, Amin et al. [26] optimize antibodies by sampling from a LLM trained on clonal families, using a twisted SMC procedure to incorporate knowledge about previous experimental measurements. A detailed comparison of ProSpero and Amin et al. [26] can be found in Appendix E.

**Generative and Energy-Based Models**  Frey et al. [27] use Langevin Markov-Chain Monte Carlo to sample from smoothed data distributions for antibody discovery. Kirjner et al. [28] construct a smoothed version of the fitness landscape prior to training a surrogate model, whose gradients are then used to guide the design of new sequences. However, the approach depends on a smoothing-strength hyperparameter, which is highly sensitive to the underlying fitness landscape and difficult to tune in practice. Frameworks introduced by Brookes et al. [14] or Song and Li [29] can propose new candidate sequences by sampling from generative models like VAEs [30]. Ghaffari et al. [31] present a VAE with a fitness-structured latent space, enabling robust optimization despite the sparsity and ruggedness of the underlying fitness landscape.

## 3 Problem formulation

We aim to discover protein sequences $x_{1:L} \in \mathcal{A}^L$ with high fitness $y \in \mathbb{R}$, where $\mathcal{A}$ denotes the vocabulary of 20 natural amino acids and $L$ represents the length of the sequence. Unless emphasis on sequential structure is needed, we simply write $x$ for brevity. The fitness value $y$ measures a given property of a protein, such as binding affinity or fluorescence intensity. Designed sequences are evaluated by a black-box oracle $f : \mathcal{A}^L \to \mathbb{R}$. Since oracle evaluations—such as wet-lab experiments or costly computational simulations—are expensive and time-consuming, queries are limited to a batch of $K$ sequences per a small number of rounds $N$. We assume access to an initial dataset $\mathcal{D}_0 = \{(x^{(i)}, y^{(i)})\}_{i=1}^M$ which allows to train a cheap and fast to query surrogate model $f_\theta : \mathcal{A}^L \to \mathbb{R}$, approximating the oracle. Additionally, let $x_{\text{start}} = \arg\max_{x \in \mathcal{D}_0} y$ correspond to the wild-type sequence. The main goal is to design protein sequences with high fitness values assigned by the oracle across $N$ active learning iterations. Desirably, generated sequences should also be biologically plausible, novel, and diverse.

# 4 Proposed framework

---

**Algorithm 1:** Active Learning with PROSPERO

---

**Input:** Oracle $f$, proxy $f_\theta$, initial dataset $\mathcal{D}_0$, active learning rounds $N$, pre-trained generative model $\mathcal{P}$, oracle budget $K$, SMC batch size $B$

1 **for** $n \leftarrow 1$ **to** $N$ **do**
2      Fit $f_\theta$ on dataset $\mathcal{D}_{n-1}$:
      $\theta \leftarrow \arg\min_\theta \mathbb{E}_{x \sim D_{n-1}} \left[ (f(x) - f_\theta(x))^2 \right]$
3      Select starting sequence:
      $x_{\text{start}} \leftarrow \arg\max_{x \in \mathcal{D}_{n-1}} f(x)$
4      Get masked variants of $x_{\text{start}}$:
      $\left\{ \tilde{x}^{(i)} \right\}_{i=1}^{B} \leftarrow$ TARGETEDMASKING$(x_{\text{start}}, f_\theta)$          // Algorithm 2
5      Propose new candidate sequences:
      $\left\{ x^{(i)} \right\}_{i=1}^{K} \leftarrow$ CONSTRAINEDSMC $\left( \left\{ \tilde{x}^{(i)} \right\}_{i=1}^{B}, \mathcal{P}, f_\theta \right)$      // Algorithm 3
6      Evaluate candidates with the oracle:
      $\hat{\mathcal{D}}_n \leftarrow \left\{ (x^{(i)}, f(x^{(i)})) \right\}_{i=1}^{K}$
7      Update dataset:
      $\mathcal{D}_n \leftarrow \mathcal{D}_{n-1} \cup \hat{\mathcal{D}}_n$

---

PROSPERO follows the active learning loop illustrated in Figure 1 and outlined in Algorithm 1, consisting of: (i) training the surrogate model $f_\theta$ on the current dataset $\mathcal{D}_{n-1}$ by minimizing the loss $\mathcal{L}(\theta) = \mathbb{E}_{x \sim D_{n-1}} \left[ (f(x) - f_\theta(x))^2 \right]$; (ii) identifying and masking fitness-relevant residues in $x_{\text{start}}$ with *targeted masking*; (iii) sampling new candidate sequences using *biologically-constrained SMC*; (iv) evaluating candidates with the oracle $f$ and augmenting $\mathcal{D}_{n-1}$. As demonstrated by Antoniuk et al. [32], the use of the active learning loop is expected to expand the support of the surrogate model, thereby providing a more reliable guiding signal to the pre-trained generative model. In the remainder of this section, we describe two core innovations of our framework: the targeted masking and the biologically-constrained SMC (line 4 and line 5 of Algorithm 1, respectively), which represent the key methodological advances of PROSPERO over prior approaches.

## 4.1 Targeted masking

Our targeted masking strategy is inspired by alanine scanning, a mutagenesis technique used both experimentally and *in silico* to identify functionally important residues by substituting each position with alanine—a neutral amino acid that disrupts side-chain interactions [33–36]. Traditionally, alanine scanning aims to locate critical residues one at a time based on wet-lab experiments or *in silico* structural modeling. In PROSPERO, we propose a batched strategy that operates purely in sequence space to identify positions within $x_{\text{start}}$ that are fitness-relevant but at the same time tolerant to mutation. Specifically, we construct $S$ batches of $B$ mutated sequences (denoted collectively as $\{x^{(i)}\}_{i=1}^{B \cdot S}$) by randomly substituting a subset of residues at locations $\mathcal{I}^{(i)} \subset \{1, \ldots, L\}$ with alanine. Each such mutated sequence is scored by the surrogate model $f_\theta$, which returns a predictive mean and uncertainty estimate: $f_\theta(x^{(i)}) = (\mu_\theta(x^{(i)}), \sigma_\theta(x^{(i)}))$. We select the top $B$ sequences according to the Upper Confidence Bound (UCB) [37] acquisition function, identifying substitutions that are not immediately harmful yet exhibit uncertainty suggestive of functional relevance of the affected residues. For each selected sequence, we construct a partially masked sequence $\tilde{x}^{(i)}$ by replacing previously substituted positions with a mask token: $\tilde{x}^{(i)}[j] = $ [MASK] if $j \in \mathcal{I}^{(i)}$, and $\tilde{x}^{(i)}[j] = x_{\text{start}}[j]$ otherwise. The resulting batch $\{\tilde{x}^{(i)}\}_{i=1}^{B}$ is then used as input to the guided generative procedure described next.

## 4.2 Biologically-constrained Sequential Monte Carlo

To design sequences with high fitness, one aims to sample from the posterior $p(x \mid y) \propto p(y \mid x)\, p(x)$. Since querying the true fitness function given by an oracle $f$ is assumed to be expensive, $p(y \mid x)$ is

typically approximated using a surrogate $f_\theta$. This yields the following target posterior distribution:

$$\gamma(x) = \frac{f_\theta(x) \cdot \mathcal{P}(x)}{Z}, \tag{1}$$

where $\mathcal{P}(x)$ denotes a prior over sequences and $Z$ is a normalization constant. $\mathcal{P}(x)$ can be modeled in various ways; here, we use EvoDiff-OADM [38]. This formulation poses two key challenges: (i) surrogate models may exhibit low fidelity on out-of-distribution sequences, and (ii) direct sampling from $\gamma(x)$ is infeasible due to the intractability of $Z$. Next, we describe how PROSPERO overcomes these challenges.

**Addressing surrogate misspecification with biologically constrained exploration**    To encourage biologically plausible exploration even under potential surrogate misspecification, we constrain candidate sampling in PROSPERO by leveraging the *charge class* of wild-type residues as an explicit biological prior. This draws inspiration from reduced amino acid alphabets (RAAs), which simplify sequence space by grouping residues with similar physicochemical and functional properties, exploiting the many-to-one relationship between sequence and structure [39, 40]. Biasing exploration toward substitutions within the same class favors alternative sequences that are more likely to preserve wild-type fitness, irrespective of the surrogate quality. We select charge as the grouping criterion, given its both fundamental and universal role in stabilizing protein structure via salt bridges, hydrogen bonding, and electrostatic interactions [41]. Specifically, for each partially masked sequence $\tilde{x}^{(i)}$, we further divide the set of masked positions $\mathcal{I}^{(i)}$ into three disjoint subsets: positive $\mathcal{I}_{(+)}^{(i)} = \{j \in \mathcal{I}^{(i)} \mid x_{\text{start}}[j] \in \{R, K, H\}\}$; negative $\mathcal{I}_{(-)}^{(i)} = \{j \in \mathcal{I}^{(i)} \mid x_{\text{start}}[j] \in \{D, E\}\}$; and neutral $\mathcal{I}_{(0)}^{(i)} = \{j \in \mathcal{I}^{(i)} \mid x_{\text{start}}[j] \notin \{R, K, H, D, E\}\}$. We formalize constrained sampling as sampling from the conditional distribution $\mathcal{P}_{RAA}(\cdot \mid \cdot)$, defined over normalized logits of the base model $\mathcal{P}$ restricted to amino acids in the same charge class as the wild-type residues at positions $\mathcal{I}^{(i)}$.

**Sampling from an intractable distribution using Sequential Monte Carlo**    To sample from the intractable target distribution $\gamma(x)$, in PROSPERO we perform approximate inference using SMC. Rather than directly sampling from complex, high-dimensional $\gamma(x_{1:L})$ defined over full sequences, SMC decomposes the problem into sequential sampling from a series of simpler, unnormalized intermediate target distributions $\{\tilde{\gamma}_l(x_{1:l})\}_{l=1}^L$, relying on a tractable proposal distribution and resampling based on intermediate importance weights (for background, refer to Appendix C). This allows us to sample sequences from the approximate target posterior in a residue-by-residue manner. We start by capitalizing on EvoDiff's order-agnostic nature by defining the sampling permutation order for each $\tilde{x}^{(i)}$ as $\pi^{(i)} = \text{concat}(\{j \notin \mathcal{I}^{(i)}\}, \mathcal{I}_{(-)}^{(i)}, \mathcal{I}_{(+)}^{(i)}, \mathcal{I}_{(0)}^{(i)})$. For simplicity, we assume that all sequences in the batch share the same number of masked positions $|\mathcal{I}| = \max_{i=1}^B |\mathcal{I}^{(i)}|$. In practice, for sequences with fewer masked tokens, no new proposals are made once all masked positions have been filled, but these sequences remain in the population and are still included in weighting and resampling (see Algorithm 3). We proceed by performing the following operations at each unmasking step $t = L - |\mathcal{I}| + 1, \ldots, L$:

(i) **Constrained proposal**: for each $\tilde{x}^{(i)}$, we sample an amino acid at position $\pi^{(i)}(t)$ from the constrained base model: $\tilde{x}_{\pi(t)}^{(i)} \sim \mathcal{P}_{RAA}(\tilde{x}_{\pi(t)}^{(i)} \mid \tilde{x}_{\pi(<t)}^{(i)})$.

(ii) **Weighting**: since the surrogate model $f_\theta$ operates only on fully unmasked sequences, intermediate targets $\tilde{\gamma}_t(\tilde{x}_{\pi(\leq t)})$ are defined only implicitly and cannot be used directly to compute importance weights $w_t$. Instead, we approximate $w_t$ by first rolling out the remainder of each sequence with the base model:

$$x_{\text{unroll}}^{(i)} \sim \prod_{s=t+1}^{T} \mathcal{P}_{RAA}(\tilde{x}_{\pi(s)}^{(i)} \mid \tilde{x}_{\pi(<s)}^{(i)}), \tag{2}$$

followed by scoring it with the surrogate model $\hat{y}^{(i)} = \mu_\theta(x_{\text{unroll}}^{(i)}) + k \cdot \sigma_\theta(x_{\text{unroll}}^{(i)})$. As the logits of the base model are constrained to charge-compatible amino acids, samples from $\mathcal{P}_{RAA}$ do not reflect the true prior over sequences. We therefore compute the perplexity of the unconstrained model $\mathcal{P}$, compensating for cases where $\mathcal{P}_{RAA}$ assigns uniformly low

likelihoods across all available choices:

$$\text{Perp}(x^{(i)}_{\text{unroll}}) = \exp\left(-\frac{1}{|\mathcal{I}|}\sum_{s=T-|\mathcal{I}|+1}^{T}\log\mathcal{P}(x^{(i)}_{\text{unroll}_{\pi(s)}} \mid x^{(i)}_{\text{unroll}_{\pi(<s)}})\right). \tag{3}$$

Finally, the unnormalized importance weights for each sequence $\tilde{x}^{(i)}$ are computed as $w_t^{(i)} = \hat{y}^{(i)}/\text{Perp}(x^{(i)}_{\text{unroll}})$, with the perplexity term correcting bias introduced by the constrained sampling.

(iii) **Resampling**: we sample a new population of partially masked sequences based on their normalized weights, effectively discarding sequences improbable under $\tilde{\gamma}_t(\tilde{x}_{\pi(\leq t)})$:

$$\tilde{x}^{(i)} \sim \text{Cat}\left(\{\tilde{x}^{(i)}\}_{i=1}^{B}, \left\{\frac{w_t^{(i)}}{\sum_{j=1}^{B}w_t^{(j)}}\right\}_{i=1}^{B}\right). \tag{4}$$

After all sequences have been fully unmasked, we select the top $K$ candidates for oracle evaluation by ranking both the final population and intermediate rollouts from the last $n_{\text{keep}}$ unmasking steps according to their predicted UCB scores $\hat{y}$.

## 5 Experiments

We show that PROSPERO successfully balances generation of high-fitness sequences with exploration, while maintaining both diversity (Section 5.1) and biological plausibility (Section 5.2). In Section 5.3 and Section 5.4, we demonstrate the robustness of our approach under different forms of surrogate misspecification, namely covariate shift and surrogate noise. Finally, Section 5.5 investigates the contribution of individual components of our framework to overall performance. We evaluate PROSPERO based on the following general setup, which serves as the basis for all subsequent experiments.

**Datasets and oracles**  We evaluate our method on eight diverse protein engineering tasks, details of which can be found in Appendix A.1. For the AAV landscape, we use ground-truth fitness scores provided in FLEXS [7]. For all other tasks, following Ren et al. [6], we use TAPE [42] as the oracle $f$ to simulate wet-lab experiments. Similarly to Kim et al. [12], we replace experimental measurements in each initial dataset with scores assigned by the oracle model.

**Baselines**  We compare our approach against a suite of established methods for iterative biological sequence design, covering diverse algorithmic paradigms: (i) evolutionary algorithms—AdaLead, PEX and CMA-ES [7, 6, 16]; (ii) on-policy reinforcement learning—DyNaPPO and LatProtRL [8, 10]; (iii) GFlowNets—GFN-AL and GFN-AL-$\delta$CS [9, 12]; (iv) evolutionary Bayesian Optimization (BO) [7]; (v) probabilistic framework CbAS [14]; and (vi) the machine-learning-assisted directed evolution (MLDE) approach of Tran and Hy [13]. Further details are provided in Appendix A.4.

**Implementation details**  We employ PROSPERO with $N = 10$ active learning rounds, each ending with a batch of $K = 128$ sequences evaluated by the oracle $f$. To model the proxy $f_\theta$, we follow Sinai et al. [7] and Kim et al. [12], and use an ensemble of three one-dimensional convolutional neural networks. This architecture is shared across all baselines to ensure a fair comparison. Sequence selection is guided by the UCB acquisition function $f_\theta(x) = \mu_\theta(x) + k \cdot \sigma_\theta(x)$, where $k = 1$ for the targeted masking and $k = 0.1$ for the biologically-constrained SMC. The SMC batch size is set to $B = 256$, and during generation we retain rollouts from the last $n_{\text{keep}} = 10$ unmasking steps. In the targeted masking, the number of alanine scans $S = 16$, with 3–10 mutations for shorter sequences ($L \approx 100$) as in the AAV, E4B, and Pab1 landscapes, and 5–15 for longer ones.

**Evaluation metrics**  We evaluate performance of all methods using four primary metrics, with particular emphasis on (i) *maximum fitness*, defined as the highest fitness value achieved among all generated sequences. This reflects the primary objective of discovering high-performing candidates. We also report: (ii) *mean fitness* of the top 100 generated sequences; (iii) *novelty*, measured as the average Hamming distance between the top 100 sequences and the starting sequence $x_{\text{start}}$; and (iv) *diversity*, defined as the average pairwise Hamming distance among the top 100 sequences.

Table 1: Maximum fitness values achieved by each method. Reported values are the mean and standard deviation over 5 runs. Green denotes fitness improvement over wild-type $x_{\text{start}}$. **Bold:** the best overall fitness per each task. Underline: second-best. PROSPERO improves fitness on every task, ranking first on 5 out of 8 and second on the remaining 3.

| Method | AMIE | TEM | E4B | Pab1 | AAV | GFP | UBE2I | LGK |
|---|---|---|---|---|---|---|---|---|
| CMA-ES | -6.857 ± 0.257 | 0.037 ± 0.01 | -0.429 ± 0.252 | 0.553 ± 0.038 | 0.000 ± 0.000 | 1.972 ± 0.135 | 0.135 ± 0.178 | -1.337 ± 0.021 |
| DyNaPPO | -3.683 ± 0.575 | 0.067 ± 0.008 | 3.924 ± 0.883 | 0.783 ± 0.036 | 0.009 ± 0.018 | 3.550 ± 0.012 | 2.796 ± 0.059 | -0.007 ± 0.015 |
| BO | 0.168 ± 0.056 | 0.682 ± 0.369 | 7.442 ± 0.242 | 0.814 ± 0.081 | 0.667 ± 0.024 | 3.584 ± 0.007 | 2.883 ± 0.069 | 0.026 ± 0.003 |
| PEX | **0.248 ± 0.007** | **1.232 ± 0.000** | 8.099 ± 0.017 | 1.499 ± 0.343 | 0.665 ± 0.022 | 3.603 ± 0.003 | 2.991 ± 0.001 | 0.037 ± 0.001 |
| AdaLead | 0.235 ± 0.002 | 1.228 ± 0.002 | 8.034 ± 0.036 | **1.978 ± 0.188** | 0.683 ± 0.037 | 3.581 ± 0.003 | 2.985 ± 0.002 | 0.038 ± 0.001 |
| CbAS | -8.202 ± 0.032 | 0.019 ± 0.002 | -0.569 ± 0.092 | 0.351 ± 0.043 | 0.000 ± 0.000 | 1.858 ± 0.067 | -0.056 ± 0.003 | -1.492 ± 0.035 |
| GFN-AL | -7.853 ± 0.270 | 0.027 ± 0.020 | 0.160 ± 0.228 | 0.507 ± 0.025 | 0.000 ± 0.000 | 2.004 ± 0.022 | 0.271 ± 0.443 | -1.164 ± 0.118 |
| GFN-AL-$\delta$CS | 0.203 ± 0.005 | 0.701 ± 0.148 | 7.930 ± 0.055 | 1.297 ± 0.337 | 0.686 ± 0.021 | 3.589 ± 0.006 | 2.984 ± 0.002 | 0.033 ± 0.001 |
| LatProtRL | 0.224 ± 0.000 | 1.229 ± 0.000 | 7.902 ± 0.086 | 1.122 ± 0.152 | 0.593 ± 0.018 | 3.590 ± 0.003 | 2.983 ± 0.000 | 0.020 ± 0.000 |
| MLDE | 0.241 ± 0.003 | 1.229 ± 0.000 | 7.934 ± 0.077 | 0.896 ± 0.015 | 0.555 ± 0.000 | 3.596 ± 0.003 | 2.984 ± 0.003 | 0.038 ± 0.002 |
| PROSPERO | 0.246 ± 0.006 | 1.231 ± 0.002 | **8.114 ± 0.037** | 1.527 ± 0.254 | **0.720 ± 0.027** | **3.617 ± 0.002** | **2.993 ± 0.003** | **0.043 ± 0.002** |

## 5.1 Protein design evaluation

**Fitness optimization** PROSPERO consistently achieves superior or comparable performance to the baselines in generating candidate sequences with high fitness, irrespective of the underlying fitness landscape (Table 1). Our approach obtains the highest maximum fitness values on 5 out of 8 protein engineering tasks and ranks second on the remaining 3. Notably, among the 11 evaluated methods, only PROSPERO and PEX are able to achieve fitness improvements over wild-type $x_{\text{start}}$ across *all* landscapes, highlighting their reliability in diverse optimization scenarios. In contrast, 4 methods fail to achieve improvements on any task. We report further results only for 6 out of 11 methods that managed to improve fitness on at least half of the tasks. Results in Table 2 demonstrate that PROSPERO is consistently able to generate a broad set of high-fitness candidates, achieving the highest mean fitness among the top 100 sequences on 5 out of 8 landscapes and ranking second on 2. Notably, mean fitness fell below that of $x_{\text{start}}$ on only a single task. Additionally, Figure 2 shows that our method discovers high-fitness sequences at earlier active learning rounds than competing approaches on half of the evaluated tasks. A detailed comparison of early-round performance across all benchmarks is provided in Appendix D.2.

**Exploration and diversity** As showcased in Table 3, PROSPERO attains high-fitness solutions with substantially greater novelty compared to other leading approaches, outperforming them on 6 out of 8 tasks. Remarkably, PROSPERO frequently breaks the conventional Pareto frontier between sequence fitness and novelty, achieving levels of both that remain mutually constraining for the competing methods (Figure 3). In particular, although PEX—the second-best performing in terms of maximum fitness—is designed to exploit the local neighborhood of the wild-type, our method often matches or exceeds its fitness while achieving approximately 2 to 9 times greater novelty. Importantly, the exceptional performance of PROSPERO in both fitness and novelty does not come at the expense of diversity. As shown in Table 9 in Appendix D.1, among leading approaches, only GFN-AL-$\delta$CS exceeds our method in diversity. However, it generates lower fitness sequences across all tasks, highlighting PROSPERO's ability to maintain diversity without compromising performance.

## 5.2 Biological plausibility

**Setup** We assess biological plausibility on the E4B task, where a large reference dataset of over 80,000 sequences not included in $\mathcal{D}_0$ is available (see Appendix A.1). Following the approach

Table 2: Mean fitness of top 100 sequences generated by leading methods. Reported values are the mean and standard deviation over 5 runs. Green: fitness improvement over wild-type $x_{\text{start}}$. **Bold:** the best overall fitness per each task. Underline: second-best. PROSPERO improves fitness on 7 out of 8 tasks, ranking first on 5 and second on 2.

| Method | AMIE | TEM | E4B | Pab1 | AAV | GFP | UBE2I | LGK |
|---|---|---|---|---|---|---|---|---|
| PEX | **0.238 ± 0.004** | **1.227 ± 0.002** | 7.948 ± 0.046 | 1.307 ± 0.258 | 0.620 ± 0.017 | 3.597 ± 0.003 | **2.987 ± 0.001** | 0.033 ± 0.001 |
| AdaLead | 0.229 ± 0.001 | 1.201 ± 0.002 | 7.846 ± 0.040 | **1.836 ± 0.266** | 0.644 ± 0.031 | 3.563 ± 0.007 | 2.976 ± 0.003 | 0.037 ± 0.001 |
| GFN-AL-$\delta$CS | -0.244 ± 0.137 | 0.192 ± 0.027 | 7.653 ± 0.136 | 1.070 ± 0.113 | 0.648 ± 0.020 | 3.569 ± 0.009 | 2.968 ± 0.006 | 0.024 ± 0.004 |
| LatProtRL | 0.217 ± 0.001 | 1.222 ± 0.000 | 7.562 ± 0.06 | 0.888 ± 0.072 | 0.563 ± 0.009 | 3.582 ± 0.003 | 2.975 ± 0.001 | 0.019 ± 0.000 |
| MLDE | 0.231 ± 0.004 | 1.131 ± 0.021 | 7.843 ± 0.122 | 0.877 ± 0.024 | 0.555 ± 0.000 | 3.591 ± 0.003 | 2.975 ± 0.005 | 0.036 ± 0.002 |
| PROSPERO | 0.236 ± 0.007 | 1.176 ± 0.029 | **8.017 ± 0.054** | 1.401 ± 0.202 | **0.679 ± 0.025** | **3.613 ± 0.002** | 2.987 ± 0.003 | **0.040 ± 0.002** |

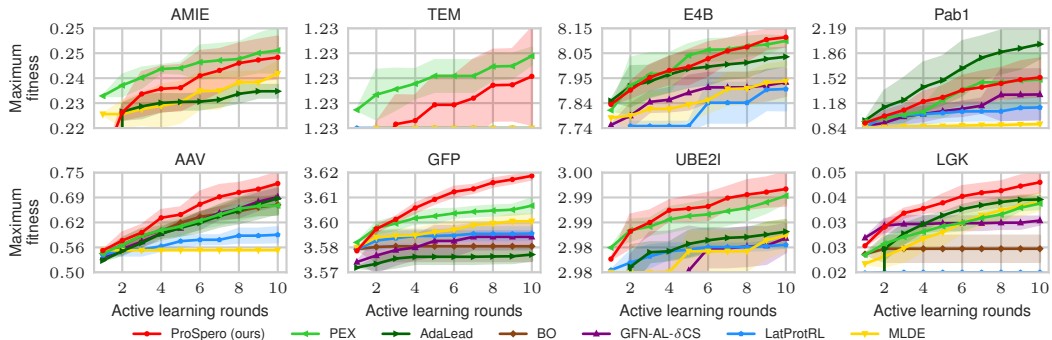

Figure 2: Maximum fitness recovered over 10 active learning rounds. Only methods that improved over $x_{\text{start}}$ are shown. Shaded regions indicate standard deviation across 5 runs. PROSPERO retrieves high-fitness sequences in earlier rounds than baselines on 4 out of 8 tasks.

Table 3: Average novelty of top 100 sequences generated by leading methods. Reported values are the mean and standard deviation over 5 runs. **Bold:** the best overall novelty. Underline: second-best. PROSPERO ranks first on 6 out of 8 tasks and second on 1.

| Method | AMIE | TEM | E4B | Pab1 | AAV | GFP | UBE2I | LGK |
|--------|------|-----|-----|------|-----|-----|-------|-----|
| PEX | $5.19 \pm 1.08$ | $1.79 \pm 0.21$ | $3.96 \pm 0.54$ | $5.29 \pm 0.74$ | $7.29 \pm 1.38$ | $6.23 \pm 1.34$ | $4.55 \pm 0.82$ | $8.45 \pm 1.39$ |
| AdaLead | $4.21 \pm 0.89$ | $2.93 \pm 0.07$ | $3.92 \pm 0.62$ | $8.47 \pm 1.58$ | $\underline{9.86 \pm 1.55}$ | $33.79 \pm 5.84$ | $3.92 \pm 0.47$ | $14.75 \pm 8.02$ |
| GFN-AL-$\delta$CS | $8.29 \pm 0.42$ | $\mathbf{10.10 \pm 0.65}$ | $4.94 \pm 0.59$ | $\underline{10.29 \pm 1.29}$ | $9.70 \pm 0.33$ | $\underline{34.83 \pm 3.77}$ | $8.01 \pm 3.69$ | $\underline{63.16 \pm 4.24}$ |
| LatProtRL | $1.09 \pm 0.04$ | $1.10 \pm 0.01$ | $3.11 \pm 0.49$ | $3.85 \pm 1.19$ | $3.03 \pm 0.40$ | $12.73 \pm 2.21$ | $1.69 \pm 0.05$ | $1.71 \pm 0.01$ |
| MLDE | $\underline{19.02 \pm 2.69}$ | $\underline{4.28 \pm 0.55}$ | $\mathbf{11.88 \pm 3.49}$ | $9.67 \pm 3.12$ | $4.48 \pm 0.25$ | $23.65 \pm 5.40$ | $\underline{9.60 \pm 2.58}$ | $50.09 \pm 8.08$ |
| PROSPERO | $\mathbf{20.99 \pm 3.32}$ | $3.37 \pm 0.53$ | $\underline{8.81 \pm 0.98}$ | $\mathbf{11.83 \pm 3.52}$ | $\mathbf{15.03 \pm 1.59}$ | $\mathbf{39.85 \pm 3.95}$ | $\mathbf{16.45 \pm 5.11}$ | $\mathbf{74.33 \pm 7.75}$ |

of Surana et al. [11], we define *validity* as the percentage of top 100 generated sequences whose key physicochemical properties fall within the central 99% quantiles of the corresponding property distributions in the reference set. Additionally, for landscapes where $x_{\text{start}}$ folds reliably, we further assess structural quality of generated sequences using pTM and pLDDT scores from ESMFold [19], as well as scPerplexity [38] computed after inverse folding with ESM-IF1 [43]. More information about the metrics is provided in Appendix A.5.

**Results** Figure 4A demonstrates that PROSPERO maintains biological plausibility while proposing sequences that are over twice as novel as those generated by baselines with comparable validity, such as PEX and LatProtRL. Moreover, it achieves this while also attaining higher fitness. Sequences generated by PROSPERO exhibit strong folding confidence, with pTM and pLDDT scores consistently above 70 (Figure 4B). Notably, this performance remains comparable to that of the wild-type even on the LGK landscape, where candidate sequences differ from the wild-type by over 70 amino acids.

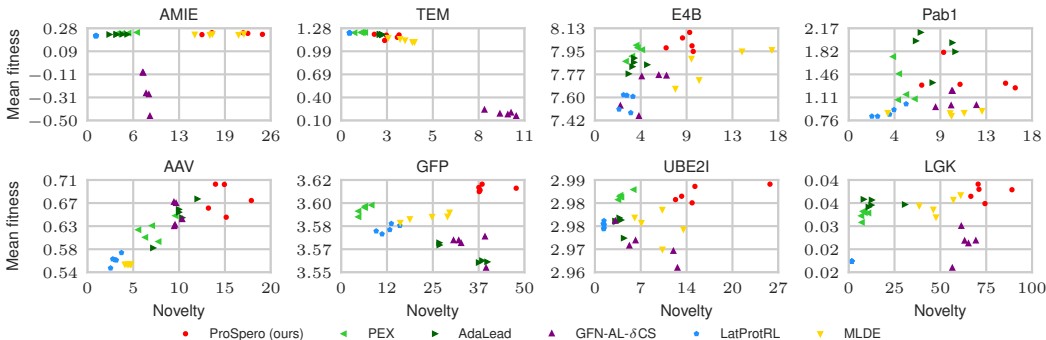

Figure 3: Comparison of fitness-novelty trade-offs among leading methods. Each dot represents the outcome of a single run. PROSPERO achieves both higher fitness and novelty than the baselines.

## 5.3 Out-of-distribution robustness

**Setup** For the simulation of distribution shifts, we used UBE2I variants and predicted their pTM scores with ESMFold [19] as the oracle. The surrogate model was trained only on sequences close to the wild-type, and optimization was then initiated from increasingly distant starting sequences $x_{\text{start}}$. We considered three cases: (i) a moderate covariate shift with $x_{\text{start}}$ differing by 35 mutations from the wild-type; (ii) a severe covariate shift with $x_{\text{start}}$ differing by 75 mutations (approximately half of the sequence length); (iii) low-data regime, where the 35-mutation case was repeated with the surrogate trained only on a small subset of the available data. Full details are provided in Appendix A.2.

**Results** Across all settings, PROSPERO consistently outperforms competing approaches, generating sequences with the highest pTM scores (Table 4), while simultaneously exploring more novel regions of the sequence space (Appendix D.3). The performance advantage over the baselines was especially pronounced under the severe shift, highlighting the effectiveness of our approach in the most challenging conditions. Even in the low-data regime, our method maintained strong performance, thereby demonstrating robustness to both distribution shifts and data scarcity. Taken together, these results show that PROSPERO remains effective in the challenging OOD settings, commonly faced when exploring sequence space beyond wild-type neighborhoods.

## 5.4 Noise robustness study

**Setup** We evaluate the robustness of exploration strategies to surrogate model misspecification by introducing increasingly noisy surrogates on the AAV landscape, where ground-truth fitness is available. Specifically, we replaced the surrogate $f_\theta$ with an ensemble of noisy oracles $f_\epsilon$, each defined by adding zero-mean Gaussian noise to the ground-truth oracle and truncating negative outputs to zero, following the perturbation scheme of Sinai et al. [7]. The magnitude of the injected noise was determined by the Signal-to-Noise Ratio (SNR), with the noise scale given by $\sigma_{\text{noise}} = \sqrt{\text{Var}(\mathcal{D}_0) \cdot 10^{-\text{SNR}/10}}$, where $\text{Var}(\mathcal{D}_0)$ denotes the variance of fitness scores in the initial dataset. This setup introduces both stochastic noise and systematic shift, as the truncation flattens low-fitness regions and biases predictions upward, making it a strong test of robustness to surrogate error.

**Results** As shown in Figure 4C, PROSPERO maintains an advantage over the majority of the baselines even at low SNR levels, demonstrating strong robustness to surrogate noise. The performance gap between our method and competing approaches widens as the noise levels decrease, highlighting PROSPERO's ability to increasingly capitalize on informative signal. Notably, our method and AdaLead exhibit a sharp performance improvement earlier than other methods, suggesting greater robustness to surrogate misspecification and ability to guide exploration toward promising regions of the search space more effectively than competing approaches.

## 5.5 Ablation

**Setup** To assess the contribution of individual components in PROSPERO, we conduct ablation studies under the same noisy surrogate setting as in the robustness analysis (Section 5.4). We compare the full method to the following ablations: (i) without SMC, corresponding to sampling from EvoDiff

Table 4: Maximum and mean pTM scores of top 100 sequences generated by leading methods under distribution shifts. Reported values are the mean and standard deviation over 5 runs. **Bold:** the best overall pTM score. Underline: second-best. PROSPERO demonstrates the highest robustness to covariate shifts.

| Method | Moderate shift | | Severe shift | | Low-data shift | |
|---|---|---|---|---|---|---|
| | Max | Mean | Max | Mean | Max | Mean |
| PEX | $0.807 \pm 0.023$ | $0.760 \pm 0.012$ | $0.578 \pm 0.014$ | $0.518 \pm 0.003$ | $0.806 \pm 0.013$ | $0.752 \pm 0.005$ |
| AdaLead | $0.796 \pm 0.013$ | $0.755 \pm 0.011$ | $0.593 \pm 0.028$ | $0.526 \pm 0.007$ | $0.781 \pm 0.016$ | $0.742 \pm 0.004$ |
| GFN-AL-$\delta$CS | $0.791 \pm 0.010$ | $0.729 \pm 0.005$ | $0.630 \pm 0.024$ | $0.542 \pm 0.006$ | $0.782 \pm 0.006$ | $0.731 \pm 0.006$ |
| LatProtRL | $0.787 \pm 0.013$ | $0.743 \pm 0.003$ | $0.560 \pm 0.000$ | $0.508 \pm 0.003$ | $0.792 \pm 0.013$ | $0.743 \pm 0.001$ |
| MLDE | $\underline{0.810 \pm 0.020}$ | $0.752 \pm 0.004$ | $\underline{0.652 \pm 0.059}$ | $\underline{0.572 \pm 0.035}$ | $0.782 \pm 0.022$ | $0.735 \pm 0.025$ |
| PROSPERO | $\mathbf{0.822 \pm 0.027}$ | $\mathbf{0.777 \pm 0.020}$ | $\mathbf{0.672 \pm 0.031}$ | $\mathbf{0.599 \pm 0.014}$ | $\mathbf{0.808 \pm 0.017}$ | $\mathbf{0.763 \pm 0.017}$ |

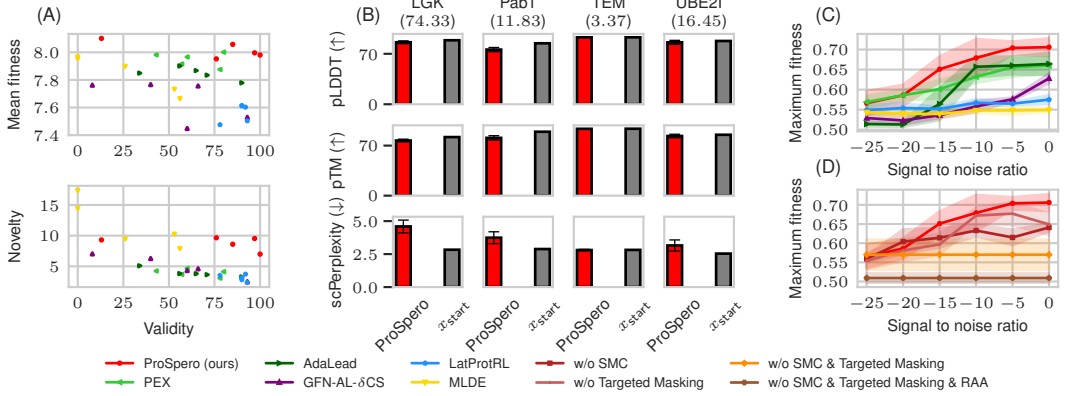

Figure 4: (A) Trade-offs between validity, fitness and novelty across leading methods; each dot represents the outcome of a single run. (B) Structural quality of top 100 sequences generated by PROSPERO across 5 runs compared to $x_{\text{start}}$; average novelty of generated sequences is shown below each task. (C) Performance of leading methods on the AAV landscape under varying levels of surrogate noise. (D) Ablation of PROSPERO components under the same setting as in (C). In both (C) and (D), shaded regions represent the standard deviation across 5 runs. PROSPERO generates highly biologically plausible sequences and remains robust to surrogate misspecification.

with the resampling steps omitted; (ii) without targeted masking, where masked positions are selected at random; (iii) without both SMC and targeted masking; and (iv) without SMC, without targeted masking, and without restricting SMC proposals to charge-compatible amino acids (without RAA).

**Results**   Results of the ablation are depicted in Figure 4D. Notably, PROSPERO with all components intact outperforms all ablated versions, with performance degrading only under extremely low SNR conditions. The advantage of our method over the baselines at low SNR levels, as seen in the noise robustness study in Section 5.4, is most likely supported by PROSPERO's constraint that limits candidate generation to sequences containing residues of the same charge class as their wild-type counterparts. This steers the generation process toward sequences with wild-type fitness regardless of the surrogate quality, providing substantial performance gains. The sharp fitness improvement at higher SNR levels likely reflects the increasing effectiveness of the SMC guidance and the targeted masking as the surrogate signal improves, while at very low SNR levels, guidance appears to slightly hinder the performance. Further ablations are provided in Appendix F.

## 6   Conclusion

In this paper, we introduced PROSPERO, an active learning framework for iterative protein sequence design based on inference-time guidance of a pre-trained generative model. Our targeted masking strategy enables edits focused on fitness-relevant residues while preserving functionally critical sites. Biologically-constrained SMC sampling allows incorporating biological prior knowledge while traversing fitness landscapes, increasing the likelihood of retrieving high-fitness sequences even under surrogate misspecification. By combining these innovations, PROSPERO enables robust exploration beyond wild-type neighborhoods while maintaining biological plausibility, achieving performance that matches or exceeds state-of-the-art approaches across diverse protein engineering tasks.

## Acknowledgments and Disclosure of Funding

We thank Rasmus Møller-Larsen, James Odgers, and Adam Izdebski for their valuable feedback and suggestions that helped us improve the manuscript. This project has received funding from the European Research Council (ERC) under the European Funding Union's Horizon 2020 research and innovation programme (grant agreement No 810115 – DOG-AMP). VF was supported by the Branco Weiss Fellowship.

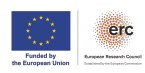

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

# Appendix

# A   Experimental details

## A.1   Protein design tasks

We evaluated PROSPERO across eight diverse protein fitness landscapes. Among these, AAV and GFP were created by Kim et al. [12] (Apache-2.0 license), while the remaining datasets were collected by Tran and Hy [13] (GPL-3.0 license). For AAV and GFP, we used oracles available in FLEXS [7] (Apache-2.0 license), while for the remaining datasets oracles provided by Ren et al. [6] (Apache-2.0 license).

(i) **Aliphatic Amide Hydrolase (AMIE).** The objective is to optimize amidase sequences for high enzymatic activity [44]. The initial dataset $\mathcal{D}_0$ includes 6417 sequences of length $L = 341$, making the search space span across $20^{341}$ possible variants. The fitness of the starting sequence is $f(x_{\text{start}}) = 0.224$, while the average distance between the starting sequence and $\mathcal{D}_0$ is $\text{Novelty}(\mathcal{D}_0, x_{\text{start}}) = 2$.

(ii) **TEM-1 $\beta$-Lactamase (TEM).** The goal is to identify TEM-1 $\beta$-lactamase variants with improved thermodynamic stability [45]. $\mathcal{D}_0$ consists of 5199 sequences with $L = 286$, $f(x_{\text{start}}) = 1.229$, and $\text{Novelty}(\mathcal{D}_0, x_{\text{start}}) = 2$.

(iii) **Ubiquitination Factor Ube4b (E4B).** The goal is to enhance the activity of the E4B ubiquitination enzyme [46]. The dataset consists of 91,032 sequences with $L = 102$, from which we randomly select 10,000 for the initial dataset $\mathcal{D}_0$. The fitness of the starting sequence $f(x_{\text{start}}) = 7.743$, with $\text{Novelty}(\mathcal{D}_0, x_{\text{start}}) = 5.42$.

(iv) **Poly(A)-binding Protein (Pab1).** The aim is to improve the binding fitness of Pab1 variants in the RNA recognition motif region [47]. The dataset contains 36,389 mutants of length $L = 75$. Similarly to E4B, we restrict $\mathcal{D}_0$ to 10,000 randomly selected sequences. The fitness of the starting sequence $f(x_{\text{start}}) = 0.843$, with $\text{Novelty}(\mathcal{D}_0, x_{\text{start}}) = 3.95$.

(v) **Adeno-associated Viruses (AAV).** The objective is to discover VP1 protein sequence fragments(positions 450–540) with improved gene therapy efficiency [48]. The dataset $\mathcal{D}_0$, of size 15,307, was created by Kim et al. [12] through random mutations of the wild-type and scoring with the oracle. Here, $L = 90$, $f(x_{\text{start}}) = 0.500$, and $\text{Novelty}(\mathcal{D}_0, x_{\text{start}}) = 5.05$.

(vi) **Green Fluorescent Proteins (GFP).** The goal is to identify protein sequences with high log-fluorescence intensity [49]. $\mathcal{D}_0$ includes 10200 sequences mutated by Kim et al. [12], as with AAV. In this case, $L = 238$, $f(x_{\text{start}}) = 3.572$, and $\text{Novelty}(\mathcal{D}_0, x_{\text{start}}) = 42.87$.

(vii) **SUMO E2 conjugase (UBE2I).** The aim here is to optimize variants of SUMO E2 conjugase for functional mapping applications [50]. $\mathcal{D}_0$ consists of 3022 sequences with $L = 159$, $f(x_{\text{start}}) = 2.978$, and $\text{Novelty}(\mathcal{D}_0, x_{\text{start}}) = 2$.

(viii) **Levoglucosan Kinase (LGK).** The objective is to optimize levoglucosan kinase variants for improved enzymatic activity [51]. $\mathcal{D}_0$ contains 7633 sequences with $L = 439$, $f(x_{\text{start}}) = 0.020$, and $\text{Novelty}(\mathcal{D}_0, x_{\text{start}}) = 2$.

## A.2   Out-of-distribution robustness

In this experiment, we used candidate sequences generated by different exploration methods during UBE2I optimization task in Section 5.1. We employed ESMFold [19] as the oracle to predict pTM scores, as protein structure prediction models provide highly reliable feedback, better suited for assessing performance in challenging OOD settings [52]. From all UBE2I candidates, we selected those within a Hamming distance $\leq 5$ from the wild-type to construct the initial dataset $\mathcal{D}_0$, consisting of 2624 sequence–pTM pairs, and with the average pTM score of $0.860 \pm 0.029$. The surrogate model $f_\theta$ was trained on $\mathcal{D}_0$, and optimization was performed for 4 iterations starting from sequences increasingly distant from the wild-type. We considered three cases:

(i) **Moderate covariate shift.** Optimization was initiated from a starting sequence $x_{\text{start}}$ differing by 35 mutations from the wild-type, and with an initial pTM of approximately 0.70.

(ii) **Severe covariate shift.** Optimization was initiated from a starting sequence $x_{\text{start}}$ differing by 75 mutations from the wild-type, and with an initial pTM of approximately 0.50.

(iii) **Low-data regime shift.** The moderate shift experiment was repeated with the surrogate trained on only 200 randomly selected points from $\mathcal{D}_0$, resulting in a reduced training set with an average pTM score of $0.860 \pm 0.023$.

Across all three cases, the exploration algorithms were run with their configurations corresponding to shorter sequences, as detailed in Section 5 for PROSPERO and in Appendix A.4 for the baselines.

## A.3 Surrogate training

Following Kim et al. [12], we trained surrogate models using the Adam optimizer [53], with both the learning rate and L2 penalty set to $0.0001$, and a batch size of 256. The maximum number of proxy updates was set to 3000, but we employed early stopping with 10% of the dataset reserved for validation, terminating training if the validation loss failed to improve for 10 consecutive iterations.

## A.4 Baselines implementation

For the following baselines, we employed the open-source implementations provided by the FLEXS benchmark [7], available at `https://github.com/samsinai/FLEXS/tree/master` under the Apache-2.0 license.

(i) **AdaLead** [7]: We followed the default hyperparameter settings provided by the authors. Specifically, we used a recombination rate of $0.2$, a mutation rate of $1/L$, where $L$ is the sequence length, and a threshold $\tau = 0.05$.

(ii) **DyNaPPO** [8]: We altered the implementation of DyNaPPO from FLEXS following the approach of Kim et al. [12]. Specifically, we replaced originally proposed proxy architectures with the same one-dimensional CNN ensembles used for other methods.

(iii) **CbAS** [14]: We implemented CbAS using a VAE [30] as the generator, retraining it at each cycle using top 20% of sequences weighted by the density ratio between the ground-truth-conditioned distribution and current sampling distribution.

(iv) **BO** [7]: We select starting sequences via Thompson sampling and use UCB to select local mutations based on surrogate model's predicted mean and uncertainty.

(v) **CMA-ES** [16]: Following Sinai et al. [7], we convert the continuous outputs from CMA-ES to one-hot representations by taking the argmax at each sequence position.

We adapted the implementation of **MLDE** [13] from `https://github.com/HySonLab/Directed_Evolution` under the GPL-3.0 license. We run 10 surrogate-based optimization steps with a population size of 128 and a beam size of 4. The random-to-importance masking ratio was set to 0.6:0.4, and we used ESM-2 [19] with 35 million parameters for unmasking.

For comparisons with **LatProtRL** [10], we used the code available at `https://github.com/haewonc/LatProtRL` under the MIT license. We employed a pre-trained ESM-2 [19] for both the encoder and decoder components of VED. For tasks involving shorter sequences (AAV, E4B, Pab1) we set: (i) VED latent dimension $R = 16$; (ii) action perturbation magnitude $\delta = 0.1$; (iii) episode length $T_{\text{ep}} = 4$; (iv) constrained decoding term $m_{\text{decode}} = 8$. For tasks involving longer sequences (GFP, AMIE, TEM, UBE2I, LGK), we used: (i) $R = 32$; (ii) $\delta = 0.3$; (iii) $T_{\text{ep}} = 6$; (iv) $m_{\text{decode}} = 18$.

In our GFlowNet setup, we employed the implementation available at `https://github.com/hyeonahkimm/delta_cs` under the Apache-2.0 license [12]. For the conservative strategy **GFN-AL-$\delta$CS** proposed by Kim et al. [12], we used an adaptive $\delta$ with maximum masking radius set to 0.05 and rank-based proxy training with reweighting factor $k = 0.01$. For AAV, E4B and Pab1 we set the scaling factor $\lambda = 0.1$, whereas for GFP, AMIE, TEM, UBE2I and LGK $\lambda = 1$. For comparisons with **GFN-AL** [9] we modified the above configuration by removing rank-based proxy training and using a fixed masking radius of 1.

We implemented **PEX** [6] using the code in `https://github.com/HeliXonProtein/proximal-exploration/tree/main` under the Apache-2.0 license, with the default setting of 2 random mutations and a frontier neighbor size of 5.

### A.5 Evaluation metrics

**Protein fitness optimization metrics**    Let $\mathcal{D}_{\text{best}} = \{(x^{(i)}, f(x^{(i)}))\}_{i=1}^{100}$ denote the set of the 100 highest-ranking sequence-fitness pairs generated across $N$ active learning rounds. The evaluation of exploration algorithms in our experiments was based on the following metrics:

(i) **Maximum fitness.** The primary evaluation criterion, representing the ability of an exploration algorithm to recover highly functional protein sequences:

$$\text{MaxFitness}(\mathcal{D}_{\text{best}}) = \max_{x \in \mathcal{D}_{\text{best}}} f(x). \tag{5}$$

(ii) **Mean fitness.** The mean fitness values of the top 100 candidate sequences:

$$\text{MeanFitness}(\mathcal{D}_{\text{best}}) = \frac{1}{|\mathcal{D}_{\text{best}}|} \sum_{x \in \mathcal{D}_{\text{best}}} f(x). \tag{6}$$

(iii) **Novelty.** The average Hamming distance between top 100 candidates and a starting sequence, characterizing the extent of divergence from the wild-type protein:

$$\text{Novelty}(\mathcal{D}_{\text{best}}, x_{\text{start}}) = \frac{1}{|\mathcal{D}_{\text{best}}|} \sum_{x \in \mathcal{D}_{\text{best}}} d(x, x_{\text{start}}). \tag{7}$$

(iv) **Diversity.** Defined as the mean pairwise Hamming distance between the top 100 candidate sequences, reflecting the exploration algorithm's ability to explore diverse regions of the fitness landscape:

$$\text{Diversity}(\mathcal{D}_{\text{best}}) = \frac{1}{|\mathcal{D}_{\text{best}}|(|\mathcal{D}_{\text{best}}| - 1)} \sum_{\substack{x,x' \in \mathcal{D}_{\text{best}} \\ x \neq x'}} d(x, x'), \tag{8}$$

**Biological plausibility measures**    To evaluate biological plausibility of candidate sequences generated by various exploration algorithms we used the following measures:

(i) **Validity.** Defined following Surana et al. [11] as a diagnostic measure to assess whether high fitness scores correspond to biologically plausible sequences. Specifically, it checks whether physicochemical properties of the top 100 candidates all fall within the 0.5th to 99.5th percentile range of the reference property distribution. This provides high-confidence indication of whether elevated fitness scores reflect genuine biological plausibility or rather result from surrogate and oracle misspecification. The considered properties were:
   - molecular weight
   - aromaticity
   - isoelectric point
   - grand average of hydropathy (GRAVY)
   - instability index

(ii) **pLDDT** and **pTM**. Both pLDDT and pTM are structure confidence scores predicted by ESMFold, scaled between 0 and 100 [19]. pLDDT measures the local per-residue confidence in structural accuracy, while pTM reflects the predicted global topology confidence. In both cases, values greater than 70 are indicative of high-confidence predictions.

(iii) **self-consistency Perplexity (scPerplexity)**. scPerplexity [38] quantifies how well a generated sequence can be recovered from its predicted structure. Specifically, it is defined as the negative log-likelihood of the original sequence conditioned on the structure predicted by a folding model. Lower scPerplexity values indicate that the sequence is more plausible under the inverse folding model given its predicted structure. For sequence folding we used ESMFold [19]; for inverse folding we used ESM-IF1 [43]

### A.6 Evodiff

To model the prior over protein sequences in PROSPERO we used EvoDiff-OADM with 38 million parameters, introduced by Alamdari et al. [38] and available under the MIT license.

# B Discussion

## B.1 Limitations

PROSPERO utilizes EvoDiff as its backbone, a model built upon a ByteNet-style CNN architecture [54]. Without enforcing reproducibility the approach remains computationally efficient and exhibits reasonable runtime. However, ensuring deterministic runs with CNNs typically leads to substantially longer runtimes, representing a practical limitation. Specifically, we conducted all experiments on a NVIDIA Tesla V100 32GB GPU, with the total runtime across all tasks being approximately 3 hours under non-deterministic setting and around 30 hours when enforcing reproducibility. For the noise robustness study in Section 5.4, the total runtime across all signal-to-noise ratio levels took approximately 30 minutes under non-deterministic configuration and around 12 hours in the reproducible setting. We note, however, that (i) reproducibility is not a strict requirement for practitioners, and (ii) even under deterministic configuration, the computational cost remains negligible compared to the burden of wet-lab experiments—which PROSPERO is designed to help alleviate.

PROSPERO performs well in maintaining diversity of generated sequences, as shown in Section 5.1. However, approximate inference using Sequential Monte Carlo carries an inherent risk of reduced diversity, which could potentially arise depending on design choices and remains a possible limitation of the approach.

## B.2 Future work

An interesting direction for future work is the development of adaptive strategies for reducing the amino acid alphabet. While our charge-based grouping offers broad applicability, more tailored schemes could further enhance performance on specific proteins. Another promising extension would be to apply PROSPERO in a lab-in-the-loop setting, using experimental validation as the oracle and integrating structure-based alanine scanning into targeted masking to more effectively identify critical residues.

## B.3 Broader impact

PROSPERO can advance protein engineering for therapeutics, enzymes, and sustainable materials by improving data efficiency and reducing experimental burden associated with wet-lab screening. However, as with any general-purpose protein design tool, there is a risk of misuse for designing harmful proteins or contributing to biosecurity concerns.

# C Background

**Sequential Monte Carlo (SMC).** Sequential Monte Carlo (SMC) is a class of approximate inference methods for sampling from complex, high-dimensional target distributions $\gamma(x_{1:T})$, where $x_{1:T}$ denotes a sequence of variables or partial states [55–57]. Rather than directly sampling from $\gamma(x_{1:T})$, SMC simplifies the inference by constructing a sequence of unnormalized intermediate target distributions $\{\tilde{\gamma}_t(x_{1:t})\}_{t=1}^{T}$, that progressively approximate the target. At each step $t$, a collection of weighted samples (i.e., particles) is propagated by sampling from proposal distributions $\{q_t(x_t \mid x_{1:t-1})\}_{t=2}^{T}$, and corrected using importance weights $\{w_t(x_{1:t})\}_{t=1}^{T}$, to account for discrepancies between the proposal and the intermediate target. At the initial step $t = 1$, $N$ particles are sampled independently from the proposal distribution $q_1(x_1)$, and initial importance weights $w_1^{(n)}$ are assigned for each particle $n$:

$$x_1^{(n)} \sim q_1(x_1), \quad w_1^{(n)} = \frac{\tilde{\gamma}_1(x_1^{(n)})}{q_1(x_1^{(n)})}. \tag{9}$$

Subsequently, each step $t = 2, \dots, T$ consists of the following three operations [56]:

(i) Optional resampling:

$$x_{1:t-1}^{(n)} \sim \mathrm{Cat}\left(\left\{x_{1:t-1}^{(n)}\right\}_{n=1}^{N}, \left\{\frac{w_{t-1}^{(n)}}{\sum_{m=1}^{N} w_{t-1}^{(m)}}\right\}_{i=1}^{N}\right). \tag{10}$$

(ii) Proposing:

$$x_t^{(n)} \sim q_t(x_t^{(n)} \mid x_{1:t-1}^{(n)}). \tag{11}$$

(iii) Weighting:

$$w_t^{(n)} = \frac{\tilde{\gamma}_t(x_{1:t}^{(n)})}{\tilde{\gamma}_{t-1}(x_{1:t-1}^{(n)})q(x_t^{(n)} \mid x_{1:t-1}^{(n)})} \tag{12}$$

# D  Full results

## D.1  Protein design

Table 5: Mean fitness of top 100 sequences generated by each method. Reported values are the mean and standard deviation over 5 runs.

| Method | AMIE | TEM | E4B | Pab1 | AAV | GFP | UBE2I | LGK |
|---|---|---|---|---|---|---|---|---|
| CMA-ES | -8.317 ± 0.029 | 0.013 ± 0.000 | -1.009 ± 0.029 | 0.232 ± 0.012 | 0.000 ± 0.000 | 1.593 ± 0.008 | -0.072 ± 0.004 | -1.538 ± 0.008 |
| DynaPPO | -6.493 ± 0.155 | 0.027 ± 0.002 | 0.574 ± 0.148 | 0.481 ± 0.013 | 0.000 ± 0.000 | 2.064 ± 0.068 | 1.600 ± 0.101 | -1.020 ± 0.045 |
| BO | -0.849 ± 0.474 | 0.606 ± 0.352 | 5.909 ± 0.785 | 0.510 ± 0.047 | 0.618 ± 0.010 | 3.538 ± 0.036 | 2.695 ± 0.148 | -0.017 ± 0.020 |
| PEX | 0.238 ± 0.004 | 1.227 ± 0.002 | 7.948 ± 0.046 | 1.307 ± 0.258 | 0.620 ± 0.017 | 3.597 ± 0.003 | 2.987 ± 0.001 | 0.033 ± 0.001 |
| AdaLead | 0.229 ± 0.001 | 1.201 ± 0.002 | 7.846 ± 0.040 | 1.836 ± 0.266 | 0.644 ± 0.031 | 3.563 ± 0.007 | 2.976 ± 0.003 | 0.037 ± 0.001 |
| CbAS | -8.361 ± 0.025 | 0.010 ± 0.001 | -0.820 ± 0.068 | 0.162 ± 0.082 | 0.000 ± 0.000 | 1.666 ± 0.021 | -0.072 ± 0.003 | -1.659 ± 0.023 |
| GFN-AL | -8.268 ± 0.010 | 0.015 ± 0.001 | -0.415 ± 0.091 | 0.276 ± 0.036 | 0.000 ± 0.000 | 1.776 ± 0.009 | 0.172 ± 0.396 | -1.345 ± 0.037 |
| GFN-AL-$\delta$CS | -0.244 ± 0.137 | 0.192 ± 0.027 | 7.653 ± 0.136 | 1.070 ± 0.113 | 0.648 ± 0.020 | 3.569 ± 0.009 | 2.968 ± 0.006 | 0.024 ± 0.004 |
| LatProtRL | 0.217 ± 0.001 | 1.222 ± 0.000 | 7.562 ± 0.06 | 0.888 ± 0.072 | 0.563 ± 0.009 | 3.582 ± 0.003 | 2.975 ± 0.001 | 0.019 ± 0.000 |
| MLDE | 0.231 ± 0.004 | 1.131 ± 0.021 | 7.843 ± 0.122 | 0.877 ± 0.024 | 0.555 ± 0.000 | 3.591 ± 0.003 | 2.975 ± 0.005 | 0.036 ± 0.002 |
| PROSPERO | 0.236 ± 0.007 | 1.176 ± 0.029 | 8.017 ± 0.054 | 1.401 ± 0.202 | 0.679 ± 0.025 | 3.613 ± 0.002 | 2.987 ± 0.003 | 0.040 ± 0.002 |

Table 6: Median fitness of top 100 sequences generated by each method. Reported values are the mean and standard deviation over 5 runs.

| Method | AMIE | TEM | E4B | Pab1 | AAV | GFP | UBE2I | LGK |
|---|---|---|---|---|---|---|---|---|
| CMA-ES | -8.392 ± 0.006 | 0.011 ± 0.000 | -1.046 ± 0.027 | 0.206 ± 0.013 | 0.000 ± 0.000 | 1.578 ± 0.006 | -0.079 ± 0.002 | -1.563 ± 0.008 |
| DyNaPPO | -6.731 ± 0.125 | 0.025 ± 0.001 | 0.333 ± 0.167 | 0.465 ± 0.012 | 0.000 ± 0.000 | 1.769 ± 0.033 | 1.577 ± 0.108 | -1.198 ± 0.032 |
| BO | -0.919 ± 0.591 | 0.600 ± 0.352 | 5.779 ± 0.851 | 0.489 ± 0.043 | 0.613 ± 0.009 | 3.543 ± 0.034 | 2.681 ± 0.169 | -0.009 ± 0.028 |
| PEX | 0.237 ± 0.004 | 1.228 ± 0.018 | 7.937 ± 0.050 | 1.298 ± 0.257 | 0.616 ± 0.017 | 3.597 ± 0.003 | 2.986 ± 0.001 | 0.033 ± 0.001 |
| AdaLead | 0.228 ± 0.001 | 1.198 ± 0.001 | 7.832 ± 0.042 | 1.830 ± 0.271 | 0.641 ± 0.031 | 3.561 ± 0.007 | 2.976 ± 0.003 | 0.037 ± 0.001 |
| CbAS | -8.371 ± 0.023 | 0.009 ± 0.001 | -0.841 ± 0.065 | 0.152 ± 0.084 | 0.000 ± 0.000 | 1.655 ± 0.021 | -0.073 ± 0.004 | -1.670 ± 0.024 |
| GFN-AL | -8.287 ± 0.007 | 0.014 ± 0.001 | -0.458 ± 0.083 | 0.257 ± 0.043 | 0.000 ± 0.000 | 1.758 ± 0.009 | 0.169 ± 0.396 | -1.353 ± 0.045 |
| GFN-AL-$\delta$CS | -0.184 ± 0.150 | 0.145 ± 0.026 | 7.633 ± 0.143 | 1.064 ± 0.113 | 0.645 ± 0.020 | 3.568 ± 0.009 | 2.967 ± 0.006 | 0.024 ± 0.005 |
| LatProtRL | 0.218 ± 0.001 | 1.222 ± 0.000 | 7.523 ± 0.062 | 0.876 ± 0.070 | 0.560 ± 0.010 | 3.582 ± 0.003 | 2.975 ± 0.001 | 0.018 ± 0.000 |
| MLDE | 0.231 ± 0.004 | 1.117 ± 0.038 | 7.834 ± 0.130 | 0.874 ± 0.027 | 0.555 ± 0.000 | 3.591 ± 0.003 | 2.975 ± 0.006 | 0.036 ± 0.002 |
| PROSPERO | 0.235 ± 0.007 | 1.187 ± 0.041 | 8.013 ± 0.055 | 1.392 ± 0.200 | 0.676 ± 0.024 | 3.613 ± 0.002 | 2.987 ± 0.003 | 0.040 ± 0.002 |

Table 7: Average diversity between top 100 sequences generated by each method. Reported values are the mean and standard deviation over 5 runs.

| Method | AMIE | TEM | E4B | Pab1 | AAV | GFP | UBE2I | LGK |
|---|---|---|---|---|---|---|---|---|
| CMA-ES | 247.97 ± 1.96 | 203.40 ± 4.60 | 75.23 ± 0.83 | 49.51 ± 1.67 | 60.34 ± 0.89 | 163.73 ± 2.44 | 108.01 ± 1.72 | 316.96 ± 5.31 |
| DynaPPO | 116.31 ± 0.82 | 98.68 ± 1.83 | 28.14 ± 0.31 | 22.00 ± 0.83 | 27.96 ± 0.44 | 79.06 ± 1.63 | 49.53 ± 0.81 | 157.11 ± 7.56 |
| BO | 21.46 ± 3.10 | 29.16 ± 24.73 | 15.65 ± 3.80 | 35.82 ± 4.80 | 6.92 ± 0.29 | 34.92 ± 3.13 | 32.33 ± 7.00 | 76.97 ± 57.70 |
| PEX | 7.06 ± 0.61 | 3.35 ± 0.32 | 4.65 ± 0.36 | 4.88 ± 0.87 | 7.00 ± 1.05 | 6.66 ± 0.86 | 6.24 ± 0.69 | 8.17 ± 1.47 |
| AdaLead | 6.37 ± 0.93 | 3.96 ± 0.06 | 5.82 ± 0.41 | 3.83 ± 0.72 | 8.09 ± 2.81 | 26.58 ± 12.68 | 6.04 ± 0.62 | 6.59 ± 1.17 |
| CbAS | 236.10 ± 25.20 | 232.03 ± 5.38 | 76.31 ± 8.05 | 53.47 ± 4.69 | 54.21 ± 5.59 | 145.39 ± 48.19 | 125.98 ± 6.49 | 349.88 ± 33.82 |
| GFN-AL | 323.77 ± 0.07 | 270.79 ± 0.25 | 96.13 ± 0.66 | 70.61 ± 0.16 | 79.70 ± 7.08 | 225.07 ± 0.12 | 78.26 ± 72.16 | 423.68 ± 0.70 |
| GFN-AL-$\delta$CS | 14.68 ± 0.79 | 17.97 ± 1.27 | 5.55 ± 0.96 | 5.68 ± 1.60 | 8.33 ± 1.14 | 24.20 ± 14.47 | 8.12 ± 3.73 | 65.68 ± 10.56 |
| LatProtRL | 1.10 ± 0.07 | 1.27 ± 0.01 | 4.02 ± 0.41 | 3.84 ± 0.34 | 3.29 ± 0.29 | 19.97 ± 3.07 | 2.24 ± 0.09 | 1.70 ± 0.00 |
| MLDE | 10.10 ± 1.23 | 6.87 ± 0.96 | 4.65 ± 1.24 | 2.35 ± 1.40 | 1.71 ± 0.22 | 8.80 ± 1.62 | 8.86 ± 1.61 | 7.52 ± 2.06 |
| PROSPERO | 12.20 ± 1.15 | 5.10 ± 0.58 | 4.12 ± 0.28 | 3.55 ± 0.26 | 5.11 ± 0.58 | 9.90 ± 2.01 | 9.54 ± 2.35 | 17.25 ± 5.64 |

Table 8: Average novelty of top 100 sequences generated by each method. Reported values are the mean and standard deviation over 5 runs.

| Method | AMIE | TEM | E4B | Pab1 | AAV | GFP | UBE2I | LGK |
|---|---|---|---|---|---|---|---|---|
| CMA-ES | 169.18 ± 1.96 | 136.79 ± 4.91 | 51.75 ± 0.96 | 32.37 ± 1.81 | 40.17 ± 0.98 | 107.96 ± 2.41 | 71.26 ± 1.62 | 212.41 ± 5.63 |
| DynaPPO | 64.78 ± 0.55 | 54.97 ± 1.13 | 15.40 ± 0.20 | 12.13 ± 0.48 | 15.40 ± 0.28 | 43.88 ± 0.99 | 27.39 ± 0.51 | 91.65 ± 12.56 |
| BO | 19.27 ± 1.94 | 35.37 ± 36.86 | 14.69 ± 2.38 | 48.94 ± 16.18 | 7.19 ± 0.68 | 41.55 ± 9.56 | 35.58 ± 12.95 | 90.62 ± 62.22 |
| PEX | 5.19 ± 1.08 | 1.79 ± 0.21 | 3.96 ± 0.54 | 5.29 ± 0.74 | 7.29 ± 1.38 | 6.23 ± 1.34 | 4.55 ± 0.82 | 8.45 ± 1.39 |
| AdaLead | 4.21 ± 0.89 | 2.93 ± 0.07 | 3.92 ± 0.62 | 8.47 ± 1.58 | 9.86 ± 1.55 | 33.79 ± 5.84 | 3.92 ± 0.47 | 14.75 ± 8.02 |
| CbAS | 323.72 ± 1.16 | 271.22 ± 1.43 | 96.41 ± 0.98 | 71.56 ± 1.08 | 86.56 ± 0.48 | 225.15 ± 1.94 | 150.19 ± 0.36 | 423.30 ± 1.96 |
| GFN-AL | 323.19 ± 0.28 | 272.81 ± 0.22 | 97.49 ± 0.28 | 71.40 ± 0.04 | 84.06 ± 1.17 | 226.60 ± 0.16 | 152.03 ± 0.60 | 425.16 ± 0.62 |
| GFN-AL-$\delta$CS | 8.29 ± 0.42 | 10.10 ± 0.65 | 4.94 ± 1.59 | 10.29 ± 1.29 | 9.70 ± 0.33 | 34.83 ± 3.77 | 8.01 ± 3.69 | 63.16 ± 4.24 |
| LatProtRL | 1.09 ± 0.04 | 1.10 ± 0.01 | 3.11 ± 0.49 | 3.85 ± 1.19 | 3.03 ± 0.40 | 12.73 ± 2.21 | 1.69 ± 0.05 | 1.71 ± 0.01 |
| MLDE | 19.02 ± 2.69 | 4.28 ± 0.55 | 11.88 ± 3.49 | 9.67 ± 3.12 | 4.48 ± 0.25 | 23.65 ± 5.40 | 9.60 ± 2.58 | 50.09 ± 8.08 |
| PROSPERO | 20.99 ± 3.32 | 3.37 ± 0.53 | 8.81 ± 0.98 | 11.83 ± 3.52 | 15.03 ± 1.59 | 39.85 ± 3.95 | 16.45 ± 5.11 | 74.33 ± 7.75 |

Table 9: Average diversity between top 100 sequences generated by leading methods. Reported values are the mean and standard deviation over 5 runs. **Bold:** the best overall diversity. Underline: second-best. PROSPERO maintains a viable level of sequence diversity.

| Method | AMIE | TEM | E4B | Pab1 | AAV | GFP | UBE2I | LGK |
|---|---|---|---|---|---|---|---|---|
| PEX | 7.06 ± 0.61 | 3.35 ± 0.32 | 4.65 ± 0.36 | 4.88 ± 0.87 | 7.00 ± 1.05 | 6.66 ± 0.86 | 6.24 ± 0.69 | 8.17 ± 1.47 |
| AdaLead | 6.37 ± 0.93 | 3.96 ± 0.06 | **5.82 ± 0.41** | 3.83 ± 0.72 | 8.09 ± 2.81 | **26.58 ± 12.68** | 6.04 ± 0.62 | 6.59 ± 1.17 |
| GFN-AL-$\delta$CS | **14.68 ± 0.79** | **17.97 ± 1.27** | 5.55 ± 0.96 | **5.68 ± 1.60** | **8.33 ± 1.14** | 24.20 ± 14.47 | 8.12 ± 3.73 | **65.68 ± 10.56** |
| LatProtRL | 1.10 ± 0.07 | 1.27 ± 0.01 | 4.02 ± 0.41 | 3.84 ± 0.34 | 3.29 ± 0.29 | 19.97 ± 3.07 | 2.24 ± 0.09 | 1.70 ± 0.00 |
| MLDE | 10.10 ± 1.23 | 6.87 ± 0.96 | 4.65 ± 1.24 | 2.35 ± 1.40 | 1.71 ± 0.22 | 8.80 ± 1.62 | 8.86 ± 1.61 | 7.52 ± 2.06 |
| PROSPERO | 12.20 ± 1.15 | 5.10 ± 0.58 | 4.12 ± 0.28 | 3.55 ± 0.26 | 5.11 ± 0.58 | 9.90 ± 2.01 | **9.54 ± 2.35** | 17.25 ± 5.64 |

## D.2 Early-round protein design

Table 10: Maximum fitness of top 100 sequences generated by leading methods limited to 4 rounds. Reported values are the mean and standard deviation over 5 runs. **Bold:** the best overall fitness. Underline: second-best.

| Method | AMIE | TEM | E4B | Pab1 | AAV | GFP | UBE2I | LGK |
|---|---|---|---|---|---|---|---|---|
| PEX | **0.242 ± 0.001** | **1.231 ± 0.001** | 7.971 ± 0.078 | 1.064 ± 0.071 | 0.604 ± 0.018 | 3.597 ± 0.002 | 2.987 ± 0.002 | 0.030 ± 0.001 |
| AdaLead | 0.232 ± 0.003 | 1.227 ± 0.004 | 7.962 ± 0.071 | **1.397 ± 0.329** | 0.596 ± 0.014 | 3.580 ± 0.003 | 2.982 ± 0.002 | 0.032 ± 0.002 |
| GFN-AL-$\delta$CS | 0.160 ± 0.048 | 0.563 ± 0.119 | 7.859 ± 0.047 | 1.035 ± 0.094 | 0.596 ± 0.010 | 3.584 ± 0.005 | 2.972 ± 0.013 | 0.032 ± 0.002 |
| LatProtRL | 0.224 ± 0.000 | 1.229 ± 0.000 | 7.751 ± 0.016 | 1.031 ± 0.129 | 0.565 ± 0.011 | 3.589 ± 0.003 | 2.982 ± 0.001 | 0.020 ± 0.000 |
| MLDE | 0.231 ± 0.006 | 1.229 ± 0.000 | 7.821 ± 0.063 | 0.866 ± 0.024 | 0.555 ± 0.000 | 3.589 ± 0.003 | 2.978 ± 0.000 | 0.028 ± 0.003 |
| PROSPERO | 0.236 ± 0.007 | 1.229 ± 0.001 | **7.978 ± 0.055** | 1.202 ± 0.129 | **0.635 ± 0.019** | **3.602 ± 0.002** | **2.989 ± 0.002** | **0.036 ± 0.001** |

Table 11: Mean fitness of top 100 sequences generated by leading methods limited to 4 rounds. Reported values are the mean and standard deviation over 5 runs. **Bold:** the best overall fitness. Underline: second-best.

| Method | AMIE | TEM | E4B | Pab1 | AAV | GFP | UBE2I | LGK |
|---|---|---|---|---|---|---|---|---|
| PEX | **0.230 ± 0.001** | 1.176 ± 0.008 | 7.686 ± 0.035 | 0.891 ± 0.036 | 0.553 ± 0.012 | 3.589 ± 0.002 | **2.982 ± 0.001** | 0.026 ± 0.001 |
| AdaLead | 0.224 ± 0.002 | 1.185 ± 0.011 | 7.682 ± 0.038 | **1.118 ± 0.222** | 0.554 ± 0.010 | 3.557 ± 0.007 | 2.964 ± 0.007 | 0.029 ± 0.002 |
| GFN-AL-$\delta$CS | −0.936 ± 0.411 | 0.108 ± 0.017 | 7.271 ± 0.232 | 0.865 ± 0.065 | 0.549 ± 0.006 | 3.562 ± 0.007 | 2.772 ± 0.141 | 0.017 ± 0.005 |
| LatProtRL | 0.200 ± 0.001 | **1.213 ± 0.000** | 6.794 ± 0.125 | 0.743 ± 0.044 | 0.525 ± 0.004 | 3.571 ± 0.005 | 2.960 ± 0.006 | 0.018 ± 0.000 |
| MLDE | 0.212 ± 0.011 | 1.060 ± 0.024 | 7.538 ± 0.109 | 0.786 ± 0.059 | 0.551 ± 0.000 | 3.585 ± 0.003 | 2.911 ± 0.019 | 0.025 ± 0.004 |
| PROSPERO | 0.221 ± 0.010 | 1.014 ± 0.104 | **7.781 ± 0.048** | 1.009 ± 0.072 | **0.576 ± 0.017** | **3.595 ± 0.002** | 2.976 ± 0.006 | **0.033 ± 0.002** |

Table 12: Average diversity between top 100 sequences generated by leading methods limited to 4 rounds. Reported values are the mean and standard deviation over 5 runs. **Bold:** the best overall diversity. Underline: second-best.

| Method | AMIE | TEM | E4B | Pab1 | AAV | GFP | UBE2I | LGK |
|---|---|---|---|---|---|---|---|---|
| PEX | 5.02 ± 0.50 | 3.07 ± 0.22 | 3.55 ± 0.36 | 3.93 ± 0.46 | 5.27 ± 0.76 | 5.85 ± 0.99 | 4.61 ± 0.63 | 7.25 ± 1.29 |
| AdaLead | 4.34 ± 0.34 | 4.04 ± 0.07 | 4.19 ± 0.26 | 3.12 ± 0.50 | **7.00 ± 0.43** | **32.04 ± 10.76** | 5.38 ± 0.77 | 4.36 ± 1.29 |
| GFN-AL-$\delta$CS | **21.22 ± 5.12** | **20.39 ± 0.75** | 5.21 ± 1.47 | **5.49 ± 0.85** | 6.53 ± 0.52 | 21.13 ± 13.93 | **13.14 ± 3.75** | **62.52 ± 9.11** |
| LatProtRL | 1.60 ± 0.04 | 1.81 ± 0.01 | 4.24 ± 0.14 | 4.02 ± 0.55 | 3.34 ± 0.12 | 27.93 ± 6.33 | 2.32 ± 0.01 | 1.79 ± 0.00 |
| MLDE | 10.99 ± 1.45 | 6.93 ± 0.62 | **5.53 ± 0.75** | 3.98 ± 0.98 | 1.63 ± 0.40 | 8.06 ± 1.27 | 7.36 ± 1.88 | 8.29 ± 1.82 |
| PROSPERO | 11.84 ± 1.82 | 6.09 ± 1.19 | 4.11 ± 0.48 | 3.65 ± 0.25 | 4.88 ± 0.45 | 9.18 ± 0.88 | 9.08 ± 0.78 | 21.18 ± 2.40 |

Table 13: Average novelty of top 100 sequences generated by leading methods limited to 4 rounds. Reported values are the mean and standard deviation over 5 runs. **Bold:** the best overall novelty. Underline: second-best.

| Method | AMIE | TEM | E4B | Pab1 | AAV | GFP | UBE2I | LGK |
|---|---|---|---|---|---|---|---|---|
| PEX | $2.83 \pm 0.24$ | $1.57 \pm 0.12$ | $1.97 \pm 0.27$ | $2.33 \pm 0.41$ | $3.68 \pm 0.17$ | $4.01 \pm 0.78$ | $2.67 \pm 0.48$ | $4.45 \pm 1.00$ |
| AdaLead | $2.77 \pm 0.44$ | $3.01 \pm 0.05$ | $2.41 \pm 0.19$ | $4.12 \pm 0.89$ | $\underline{4.82 \pm 0.36}$ | $\mathbf{34.94 \pm 4.19}$ | $3.41 \pm 0.54$ | $8.30 \pm 8.37$ |
| GFN-AL-$\delta$CS | $\underline{11.68 \pm 2.71}$ | $\mathbf{11.45 \pm 0.40}$ | $3.52 \pm 1.71$ | $4.99 \pm 0.81$ | $4.52 \pm 0.26$ | $\underline{30.61 \pm 3.44}$ | $\mathbf{13.14 \pm 1.98}$ | $\mathbf{45.70 \pm 3.31}$ |
| LatProtRL | $1.36 \pm 0.034$ | $1.61 \pm 0.01$ | $3.19 \pm 0.18$ | $2.87 \pm 0.45$ | $1.88 \pm 0.06$ | $16.44 \pm 3.56$ | $1.75 \pm 0.07$ | $1.85 \pm 0.01$ |
| MLDE | $\mathbf{11.85 \pm 2.46}$ | $\underline{4.71 \pm 0.50}$ | $\mathbf{7.11 \pm 1.21}$ | $\underline{5.76 \pm 0.67}$ | $4.10 \pm 0.39$ | $13.49 \pm 1.48$ | $6.36 \pm 0.67$ | $18.90 \pm 5.66$ |
| PROSPERO | $10.67 \pm 1.62$ | $3.40 \pm 0.58$ | $\underline{3.95 \pm 0.88}$ | $\mathbf{6.16 \pm 2.36}$ | $\mathbf{6.60 \pm 0.78}$ | $17.16 \pm 3.23$ | $\underline{9.02 \pm 3.48}$ | $\underline{37.09 \pm 4.32}$ |

## D.3 Out-of-distribution robustness

Table 14: Results under moderate covariate shift. Reported values are the mean and standard deviation over 5 runs. **Bold:** the best overall value. Underline: second-best.

| Method | Maximum pTM | Mean pTM | Diversity | Novelty |
|---|---|---|---|---|
| PEX | $0.807 \pm 0.023$ | $\underline{0.760 \pm 0.012}$ | $6.14 \pm 0.89$ | $4.45 \pm 0.38$ |
| AdaLead | $0.796 \pm 0.013$ | $0.755 \pm 0.011$ | $8.83 \pm 2.54$ | $8.36 \pm 2.97$ |
| GFN-AL-$\delta$CS | $0.791 \pm 0.010$ | $0.729 \pm 0.005$ | $\mathbf{16.92 \pm 0.88}$ | $9.56 \pm 0.60$ |
| LatProtRL | $0.787 \pm 0.013$ | $0.743 \pm 0.003$ | $6.32 \pm 0.32$ | $5.90 \pm 0.53$ |
| MLDE | $\underline{0.810 \pm 0.020}$ | $0.752 \pm 0.004$ | $9.89 \pm 1.11$ | $\mathbf{20.88 \pm 2.98}$ |
| PROSPERO | $\mathbf{0.822 \pm 0.027}$ | $\mathbf{0.777 \pm 0.020}$ | $\underline{11.50 \pm 1.62}$ | $\underline{17.74 \pm 3.20}$ |

Table 15: Results under severe covariate shift. Reported values are the mean and standard deviation over 5 runs. **Bold:** the best overall value. Underline: second-best.

| Method | Maximum pTM | Mean pTM | Diversity | Novelty |
|---|---|---|---|---|
| PEX | $0.578 \pm 0.014$ | $0.518 \pm 0.003$ | $3.40 \pm 0.07$ | $1.72 \pm 0.04$ |
| AdaLead | $0.593 \pm 0.028$ | $0.526 \pm 0.007$ | $14.26 \pm 1.91$ | $7.66 \pm 1.08$ |
| GFN-AL-$\delta$CS | $0.630 \pm 0.024$ | $0.542 \pm 0.006$ | $\mathbf{24.13 \pm 1.47}$ | $14.63 \pm 1.16$ |
| LatProtRL | $0.560 \pm 0.000$ | $0.508 \pm 0.003$ | $2.24 \pm 0.14$ | $1.78 \pm 0.16$ |
| MLDE | $\underline{0.652 \pm 0.059}$ | $\underline{0.572 \pm 0.035}$ | $13.10 \pm 1.18$ | $\underline{21.68 \pm 3.85}$ |
| PROSPERO | $\mathbf{0.672 \pm 0.031}$ | $\mathbf{0.599 \pm 0.014}$ | $\underline{14.51 \pm 1.99}$ | $\mathbf{22.03 \pm 1.69}$ |

Table 16: Results under low-data covariate shift. Reported values are the mean and standard deviation over 5 runs. **Bold:** the best overall value. Underline: second-best.

| Method | Maximum pTM | Mean pTM | Diversity | Novelty |
|---|---|---|---|---|
| PEX | $\underline{0.806 \pm 0.013}$ | $\underline{0.752 \pm 0.005}$ | $6.77 \pm 0.45$ | $4.39 \pm 0.51$ |
| AdaLead | $0.781 \pm 0.016$ | $0.742 \pm 0.004$ | $7.99 \pm 1.39$ | $5.95 \pm 2.04$ |
| GFN-AL-$\delta$CS | $0.782 \pm 0.006$ | $0.731 \pm 0.006$ | $\mathbf{15.82 \pm 1.77}$ | $9.46 \pm 1.69$ |
| LatProtRL | $0.792 \pm 0.013$ | $0.743 \pm 0.001$ | $6.25 \pm 0.23$ | $5.57 \pm 0.20$ |
| MLDE | $0.782 \pm 0.022$ | $0.735 \pm 0.025$ | $9.39 \pm 2.88$ | $\mathbf{16.97 \pm 3.78}$ |
| PROSPERO | $\mathbf{0.808 \pm 0.017}$ | $\mathbf{0.763 \pm 0.017}$ | $\underline{11.25 \pm 2.51}$ | $\underline{15.87 \pm 1.17}$ |

# E Extended related work

**Inference-time guidance methods** Several methods have been proposed to steer generative models during inference. Dhariwal and Nichol [58] introduced classifier guidance, where the sampling is biased toward desired properties by adjusting the generative score with the gradient of an auxiliary classifier. However, this approach is not directly applicable in the discrete data domain, where

gradients with respect to inputs are not well-defined. To address this, Nisonoff et al. [59] developed a framework that enables classifier guidance in discrete diffusion and flow matching models. Their approach leverages a continuous-time Markov chain formulation of the forward process and corresponding reverse-time generative process [60], where only one coordinate changes at each transition, making exact guidance tractable.

Guidance can also be realized through SMC-based approaches, applicable in both discrete and continuous domains [61–63]. These methods steer generation by maintaining a population of particles that represent partial trajectories and resampling them according to their likelihood under a target distribution. Ekström Kelvinius and Lindsten [64] extend discriminator guidance [65], originally developed for score-based diffusion models, to Autoregressive Diffusion Models (ARDMs) [66] (such as leveraged in our work EvoDiff-OADM [38]), and further employ SMC to correct for discriminator errors at intermediate sampling steps. Li et al. [67] take a similar approach that resembles SMC in the use of importance sampling, but instead of resampling across the entire batch of particles, they generate and reweight multiple candidates from each individual sample at the previous step. Building on this idea, Uehara et al. [68] combine it with a noising policy, iteratively alternating between re-noising and reward-guided denoising to progressively refine samples.

**Similarities to CloneBO [26]**    PROSPERO and CloneBO by Amin et al. [26] share similarities in guiding a generative model at inference time using SMC. However, the approaches differ meaningfully in both scope and mechanism. First, in CloneBO, the generative model (CloneLM) has been trained specifically for the optimization task by fitting to a distribution of clonal families. In contrast, our method uses a general-purpose, task-agnostic pre-trained generative model, enabling effortless optimization regardless of the protein family. As for the differences in the use of SMC, in CloneBO the authors compute intermediate importance weights directly, as a likelihood ratio between the base CloneLM and a twisted variant incorporating high value sequences (i.e. sequences with experimental measurements) in the conditioned clonal family. In PROSPERO, we approximate the intermediate importance weights using the surrogate model, which serves as a tractable proxy for the true likelihood. Moreover, our biologically-constrained SMC restricts proposals to charge-compatible amino acids, making certain residues impossible to sample. In contrast, CloneBO does not enforce such constraints; although twisting reduces the probability of sampling undesirable residues, it does not eliminate them entirely.

**Connections between SMC and pseudo-marginal MCMC**    Pseudo-marginal MCMC [69] and SMC differ in the way they approximate the target distribution. In pseudo-marginal MCMC, within a single Markov chain, a single collection of samples is generated whose marginal stationary distribution is exactly the target distribution. The approximation improves with the number of steps $T$ and is exact in the limit of inifite $T$. In contrast, SMC maintains a population of $N$ samples that evolve over a sequence of $T$ intermediate distributions. The empirical distribution formed by these samples converges to the target distribution in the limit of infinite $N$.

# F    Further ablations

**Influence of the number of starting sequences on candidate generation**    We investigated how varying the number of best starting sequences $x_{\text{start}}$ affects the proposed candidates by conducting experiments on the LGK landscape, which requires generating the longest sequences ($L = 439$). The results in Table 17 demonstrate that fewer starting points drive deeper, more directed exploration, resulting in higher novelty and fitness but lower diversity. In contrast, more starting points promote broader, more diffuse exploration, increasing diversity but limiting how far any single trajectory moves from the wild-type across subsequent optimization rounds.

**Influence of reducing the amino acid alphabet**    To further analyze the isolated effect of constraining the proposals to charge-compatible amino acids, we directly compare the performance of the full PROSPERO with its ablated counterpart lacking this restriction (without RAA). We followed the setup detailed in Section 5.5. The results in Table 18 highlight the relevance of this feature, showing consistent improvements across all signal-to-noise ratio levels.

**Influence of the charge-class permutation order**    The sampling permutation order in biologically-constrained SMC was chosen to prioritize charge classes with fewer valid options (negatively charged

residues first, followed by positively charged, and finally neutral), as resolving the most constrained decisions early should help prevent suboptimal completions later in the sequence. To assess this, we conducted an ablation comparing PROSPERO with its standard permutation order (ascending) to both a reversed (descending) order and a random order. The results presented in Table 19 show that the reasoning behind this choice appears correct, though the performance benefits are modest.

Table 17: Results on the LGK landscape for varying numbers of starting sequences. Reported values are then mean and standard deviation over 5 runs. The best overall values are highlighted in **bold**.

| $x_{\text{start}}$ | Maximum fitness | Mean fitness | Diversity | Novelty |
|---|---|---|---|---|
| 1 | $\mathbf{0.043 \pm 0.002}$ | $\mathbf{0.040 \pm 0.002}$ | $17.25 \pm 5.64$ | $\mathbf{74.33 \pm 7.75}$ |
| 4 | $0.041 \pm 0.001$ | $0.039 \pm 0.001$ | $19.30 \pm 4.58$ | $70.20 \pm 6.05$ |
| 16 | $0.038 \pm 0.002$ | $0.037 \pm 0.002$ | $19.09 \pm 5.80$ | $64.86 \pm 5.44$ |
| 64 | $0.037 \pm 0.002$ | $0.034 \pm 0.001$ | $29.13 \pm 6.97$ | $56.38 \pm 7.06$ |
| 128 | $0.033 \pm 0.002$ | $0.030 \pm 0.002$ | $\mathbf{33.89 \pm 9.54}$ | $50.45 \pm 5.52$ |

Table 18: Results on the AAV landscape across different signal-to-noise ratio levels (SNR). Reported values are then mean and standard deviation over 5 runs. The best overall values are highlighted in **bold**.

| SNR level | -25 | -20 | -15 | -10 | -5 | 0 |
|---|---|---|---|---|---|---|
| PROSPERO | $\mathbf{0.566 \pm 0.030}$ | $\mathbf{0.586 \pm 0.026}$ | $\mathbf{0.651 \pm 0.032}$ | $\mathbf{0.679 \pm 0.047}$ | $\mathbf{0.704 \pm 0.016}$ | $\mathbf{0.706 \pm 0.022}$ |
| PROSPERO w/o RAA | $0.507 \pm 0.025$ | $0.555 \pm 0.040$ | $0.588 \pm 0.029$ | $0.605 \pm 0.032$ | $0.653 \pm 0.042$ | $0.666 \pm 0.029$ |

Table 19: Results on all the landscapes with different permutation orderings. Reported values are then mean and standard deviation over 5 runs. The best overall values are highlighted in **bold**.

| Ordering | AMIE | TEM | E4B | Pab1 | AAV | GFP | UBE2I | LGK |
|---|---|---|---|---|---|---|---|---|
| Ascending | $\mathbf{0.246 \pm 0.006}$ | $1.231 \pm 0.002$ | $8.114 \pm 0.037$ | $\mathbf{1.527 \pm 0.254}$ | $\mathbf{0.720 \pm 0.027}$ | $\mathbf{3.617 \pm 0.003}$ | $2.993 \pm 0.003$ | $\mathbf{0.043 \pm 0.002}$ |
| Descending | $0.244 \pm 0.005$ | $\mathbf{1.232 \pm 0.003}$ | $8.139 \pm 0.037$ | $1.363 \pm 0.141$ | $0.706 \pm 0.035$ | $3.614 \pm 0.003$ | $2.991 \pm 0.003$ | $0.041 \pm 0.003$ |
| Random | $0.243 \pm 0.005$ | $\mathbf{1.232 \pm 0.002}$ | $\mathbf{8.164 \pm 0.015}$ | $1.338 \pm 0.113$ | $0.708 \pm 0.029$ | $3.614 \pm 0.003$ | $\mathbf{2.993 \pm 0.002}$ | $0.042 \pm 0.003$ |

# G  Algorithms

---

**Algorithm 2:** Targeted Masking

---

**Input:** Starting sequence $x_{\text{start}}$, proxy $f_\theta$, SMC batch size $B$, scans $S$, min substitutions $n_{\min}$, max substitutions $n_{\max}$, exploitation-exploration coefficient $k$

**Output:** Masked sequences $\{\tilde{x}^{(i)}\}_{i=1}^{B}$, substitution locations $\{\mathcal{I}^{(i)}\}_{i=1}^{B}$

1 **for** $i \leftarrow 1$ **to** $B \times S$ **do**

2      $n_{\text{sub}}^{(i)} \sim \mathcal{U}_{[n_{\min}, n_{\max}]}$

3      $\mathcal{I}^{(i)} \sim \text{UniformSubset}([1, |x_{\text{start}}|], \, n_{\text{sub}}^{(i)})$

4      $x^{(i)} \leftarrow x_{\text{start}}$, where $x^{(i)}[j] \leftarrow \texttt{A}$ for $j \in \mathcal{I}^{(i)}$

5 $\mathcal{J} \leftarrow \arg\max_{x \in \{x^{(i)}\}_{i=1}^{B \times S}}^{B} \mu_\theta(x) + k \cdot \sigma_\theta(x)$

6 **for** $x^{(i)} \in \mathcal{J}$ **do**

7      $\tilde{x}^{(i)} \leftarrow x^{(i)}$, where $\tilde{x}^{(i)}[j] \leftarrow \texttt{[MASK]}$ for $j \in \mathcal{I}^{(i)}$

8 **return** $\{\tilde{x}^{(i)}\}_{i=1}^{B}$, $\{\mathcal{I}^{(i)}\}_{i=1}^{B}$

---

**Algorithm 3:** ConstrainedSMC

**Input:** Partially masked sequences $\{\tilde{x}^{(i)}\}_{i=1}^B$, mask locations $\{\mathcal{I}^{(i)}\}_{i=1}^B$, pre-trained generative model $\mathcal{P}$, proxy $f_\theta$, oracle budget $K$, exploitation-exploration coefficient $k$, kept rollouts threshold $n_{\text{keep}}$

**Output:** Candidate sequences $\{x^{(i)}\}_{i=1}^K$

1   RolloutBuffer $\leftarrow \{\}$

2   **for** $i \leftarrow 1$ **to** $B$ **do**

3      $\pi^{(i)} = \text{concat}(\underline{\mathcal{I}}^{(i)}, \mathcal{I}_{(-)}^{(i)}, \mathcal{I}_{(+)}^{(i)}, \mathcal{I}_{(0)}^{(i)})$

4   $T \leftarrow \arg\max_{\mathcal{I} \in \{\mathcal{I}^{(i)}\}_{i=1}^B} |\mathcal{I}|$

5   **for** $t \leftarrow 1$ **to** $T$ **do**

6      **for** $\tilde{x}^{(i)} \in \{\tilde{x}^{(i)}\}_{i=1}^B$ **do**

7         **if** $t = 1$ **then**

8           $LL^{(i)} \leftarrow 0$

9         **if** $t \leq |\mathcal{I}^{(i)}|$ **then**

10           $\tilde{x}_{\pi(t+|\underline{\mathcal{I}}^{(i)}|)}^{(i)} \sim \mathcal{P}_{\text{RAA}}(\tilde{x}_{\pi(t+|\underline{\mathcal{I}}^{(i)}|)}^{(i)} \mid \tilde{x}_{\pi(<t+|\underline{\mathcal{I}}^{(i)}|)}^{(i)})$

11           $LL^{(i)} \leftarrow LL^{(i)} + \log \mathcal{P}(\tilde{x}_{\pi(t+|\underline{\mathcal{I}}^{(i)}|)}^{(i)} \mid \tilde{x}_{\pi(<t+|\underline{\mathcal{I}}^{(i)}|)}^{(i)})$

12           $(x_{\text{unroll}}^{(i)}, LL_{\text{unroll}}^{(i)}) \leftarrow \text{ROLLOUT}(\tilde{x}^{(i)}, \pi^{(i)}, LL^{(i)}, \mathcal{P}, s = t+1)$    // Algorithm 4

13           $\hat{y}^{(i)} \leftarrow \mu_\theta(x_{\text{unroll}}^{(i)}) + k \cdot \sigma_\theta(x_{\text{unroll}}^{(i)})$

14           $\text{invPPL}^{(i)} \leftarrow \exp\left(\frac{LL_{\text{unroll}}^{(i)}}{|\mathcal{I}^{(i)}|}\right)$

15           **if** $T - t < n_{keep}$ **then**

16             RolloutBuffer $\leftarrow$ RolloutBuffer $\cup \{(x_{\text{unroll}}^{(i)}, \hat{y}^{(i)})\}$

17      $w \leftarrow \left(\frac{\hat{y}^{(i)} \cdot \text{invPPL}^{(i)}}{\sum_{j=1}^B \hat{y}_j \cdot \text{invPPL}_j}\right)_{i=1}^B$

18      **for** $i \leftarrow 1$ **to** $B$ **do**

19         Resample:

           $idx^{(i)} \sim \text{Cat}(w)$

           $\tilde{x}^{(i)} \leftarrow \tilde{x}[idx^{(i)}]$

           $\pi^{(i)} \leftarrow \pi[idx^{(i)}]$

           $LL^{(i)} \leftarrow LL[idx^{(i)}]$

20   **return** $\{x^{(i)}\}_{i=1}^K \leftarrow \arg\max_{(x,\hat{y}) \in RolloutBuffer}^K \hat{y}$

---

**Algorithm 4:** Rollout

**Input:** Partially masked sequences $\{\tilde{x}^{(i)}\}_{i=1}^B$, sampling permutations $\{\pi^{(i)}\}_{i=1}^B$, log-likelihoods $\{LL^{(i)}\}_{i=1}^B$, pre-trained generative model $\mathcal{P}$, unmasking step $s$

**Output:** Unrolled sequences $\{x_{\text{unroll}}^{(i)}\}_{i=1}^B$, log-likelihoods $\{LL_{\text{unroll}}^{(i)}\}_{i=1}^B$

1   $T \leftarrow \arg\max_{\mathcal{I} \in \{\mathcal{I}^{(i)}\}_{i=1}^B} |\mathcal{I}|$

2   **for** $t \leftarrow s$ **to** $T$ **do**

3      **for** $\tilde{x}^{(i)} \in \{\tilde{x}^{(i)}\}_{i=1}^B$ **do**

4         **if** $t \leq |\mathcal{I}^{(i)}|$ **then**

5           $\tilde{x}_{\pi(t+|\underline{\mathcal{I}}^{(i)}|)}^{(i)} \sim \mathcal{P}_{\text{RAA}}(\tilde{x}_{\pi(t+|\underline{\mathcal{I}}^{(i)}|)}^{(i)} \mid \tilde{x}_{\pi(<t+|\underline{\mathcal{I}}^{(i)}|)}^{(i)})$

6           $LL^{(i)} \leftarrow LL^{(i)} + \log \mathcal{P}(\tilde{x}_{\pi(t+|\underline{\mathcal{I}}^{(i)}|)}^{(i)} \mid \tilde{x}_{\pi(<t+|\underline{\mathcal{I}}^{(i)}|)}^{(i)})$

7   $\{x_{\text{unroll}}^{(i)}\}_{i=1}^B \leftarrow \{\tilde{x}^{(i)}\}_{i=1}^B$

8   $\{LL_{\text{unroll}}^{(i)}\}_{i=1}^B \leftarrow \{LL^{(i)}\}_{i=1}^B$

9   **return** $\{x_{unroll}^{(i)}\}_{i=1}^B, \{LL_{unroll}^{(i)}\}_{i=1}^B$

