# OpenReview forum: "ProSpero: Active Learning for Robust Protein Design Beyond Wild-Type Neighborhoods"
_NeurIPS.cc/2025/Conference — NeurIPS 2025 poster_

### Official Review · Reviewer_Dp5Q · 2025-06-30

**Clarity:** 3
**Significance:** 3
**Originality:** 3
**Rating:** 4
**Confidence:** 3

**Summary:**

ProSpero introduces an iterative protein design method that goes beyond recent work PEX – that relies on local search, using a pretrained generative model to explore beyond a protein wild-type neighborhood. The pretrained generative model is used as a prior to guide the exploration, whilst proposing biologically plausible and diverse candidates in each round.

ProSpero is evaluated in eight multi-round design campaigns, where ground truth scores are taken from the TAPE model, and the ensemble surrogate is fine-tuned in each round. The paper introduces two methodological contributions: a targeted masking strategy to identify high-value residues to mutate, and a constrained SMC sampling process.

**Questions:**

RAAs: In practice, what is the impact of using the RAA based on charge particles? Was an ablation performed where you simply sample from the full AA alphabet?

What value of n_sub was used in practice? Is this HP ablated? i.e. how many positions do you mask and the length of $\mathcal{I}^{(i)}$? and what effect does this have on the novelty and quality of generated sequences? What is the sensitivity of this HP? My intuition is that when set too high or low, the sequences will score poorly for novelty. How was the HP set?

Line 147 defines $\pi^(i)$  as the order agnostic indices, why is it concatenated in that specific order? What is the intuition for why one would want to sample from I_neg before I_pos, followed by I_netural? Did you ablate or experiment with other potential orderings?

**Ethical Concerns:**

["NO or VERY MINOR ethics concerns only"]

**Limitations:**

Yes

**Quality:**

3

**Strengths And Weaknesses:**

Strengths:
The paper is well written and organised. It was an enjoyable read. The reported results are good relative to benchmark methods, plus reproducibility is high, with open source code provided for the main ProSpero method and all baselines.
Further, efforts have been made to generate biological meaningful sequences, via constrained search (using AA charge classes), and also evaluating the generated sequences via biological validity metrics based on physicochemical properties.

Weaknesses:
Problem formulation: As far as I understand, the goal of ProSpero is to design protein sequences with high fitness values across $N$ iterations. This feels like an optimisation objective, not an active learning one - where we are trying to improve a model.
Please could you untie or explain the difference between your proposed active learning formulation, and a Bayesian optimization objective? It would appear to me that you have an optimisation objective, unless I am missing something.

On Clarity:
I am confused exactly what distribution your sampling algorithm targets, and further, under what assumptions ProSpero does indeed target that distribution. Although clearly an effective proxy, it is not obvious to me what distribution the importance weights imply.
On the underlying assumptions: I suspect the sampling scheme is closely related to the field of pseudo-marginal MCMC (see Andrieu et al. (2009)), wherein an unbiased estimate of the unnormalised posterior is used in place of the underlying unnormalised density; precise details of this connection to such work would be appreciated.
Generally, I would greatly appreciate more details/precision with respect to your SMC scheme.

On novelty:
The significance of the proposed Targeted Masking contribution appears limited. To me this feels like random AA permutation, plus a surrogate and acquisition function applied on top.

Please elucidate the connections between [26] and your proposed method.
To my understanding, CloneBO performs a similar *iterative design of protein sequences as inference-time guidance of a pre-trained generative model*, and performs the guidance *via conditioning on high value sequences*. Is there a connection between the high value sequences used to guide the tSMC samples in CloneBO and your use of a surrogate model?

[26] Alan Nawzad Amin, et. al. Bayesian optimization of antibodies informed by a generative model of evolving sequences. ICLR 2025.

It would be of benefit if the authors situate their guidance method relative to other generative guidance methods, such as classifier guidance.

---

> ### Author Rebuttal · Authors · 2025-07-30
>
> Thank you for your meaningful and relevant feedback of our work.
>
> ## W1: Problem formulation
> We agree that the ultimate objective of our framework is optimization of protein sequences, not active learning in the traditional sense of model improvement for its own sake. Rather, the latter serves as a proxy objective to achieve the former. We appreciate the opportunity to clarify this point and will revise the manuscript to reflect this more precise framing.
>
> ## W2: Clarity
> While full SMC details are provided in Appendix C, we agree that being more precise would improve clarity. In the revised version, we will incorporate the following changes:
> - In Equation 1 of Section 4.2, we will revise the notation of the target distribution from $\gamma(x)$ to $\gamma(x_{1:L})$ , and update all other occurrences of $x$ accordingly within the introductory part of Section 4.2. This change makes explicit that the target is defined over a sequence of residues, improving clarity and alignment with the sequential structure of SMC
> - We will begin the L144-167 part with the following paragraph: Rather than trying to directly sample from complex, high-dimensional target distribution $\gamma(x_{1:L})$, in ProSpero we perform approximate inference using SMC, which decomposes the sampling process into a sequence of simpler, unnormalized, intermediate target distributions $\{\tilde{\gamma}(x_{1:l})\}_{l=1}^{L}$ (for background see Appendix C). This allows us to progressively build up sequences residue by residue, while maintaining a tractable approximation of the full design objective"
> - We will conclude the line 162 as follows: “Finally, the unnormalized importance weights […] , with the perplexity correction term penalizing sequences that are improbable under the true prior, correcting bias introduced by the constrained sampling”
>
> We hope this revision will make the section clearer and more accessible to the readers.
>
> Importance weights are proportional to the likelihood, as at each step they represent the ratio between the target distribution (i.e. posterior) and the proposal (prior). Resampling particles proportionally to their weights allows us to progressively concentrate samples in high-probability regions under the posterior.
>
> As for the connections between pseudo-marginal MCMC and SMC: in pseudo-marginal MCMC, within a single Markov chain, a *single* collection of samples is generated whose marginal stationary distribution is exactly the target distribution. The approximation improves with the number of iterations $T$ and is exact in the limit of *infinite $T$*. In contrast, SMC maintains a *population* of $N$ samples that evolve over a sequence of intermediate distributions. The empirical distribution formed by these samples converges to the target distribution in the limit of *infinite $N$*.
>
> ## W3: Novelty
> The motivation behind targeted masking is a biologically motivated strategy that mirrors how experimentalists scan for functionally relevant residues. We demonstrated its effectiveness over random masking in Figure 4D in Section 5.4. For convenience, we summarized the key results in the table below:
>
> |SNR level|-25|-20|-15|-10|-5|0|
> |-|-|-|-|-|-|-|
> |ProSpero|**0.566 ± 0.03**|**0.586 ± 0.026**|**0.651 ± 0.032**|**0.679 ± 0.047**|**0.704 ± 0.016**|**0.706 ± 0.022**|
> |ProSpero w/o TM|0.554 ± 0.021|0.581 ± 0.028|0.597 ± 0.027|0.672 ± 0.021|0.677 ± 0.030|0.649 ± 0.028|
>
> Results clearly demonstrate the advantage of targeted over random masking across all noise levels. To improve the clarity of the strategy behind targeted masking, we will improve its description in Section 4.1 in the final paper.
>
> ## W4: Similarities to CloneBO by Amin et al. [1]
>
> We agree that ProSpero and CloneBO [1] share similarities in guiding a generative model at inference time using SMC. However, the approaches differ meaningfully in both scope and mechanism.
> First, in CloneBO, the generative model (CloneLM) has been trained *specifically* for the optimization task by fiting to a distribution of clonal families. In contrast, our method uses a *general-purpose, task-agnostic* pre-trained generative model, enabling effortless optimization regardless of the protein family.
> As for the differences in the use of SMC, in CloneBO the authors compute intermediate importance weights directly, as a *likelihood ratio* between the base CloneLM and a twisted variant incorporating high value sequences (i.e. sequences with experimental measurements) in the conditioned clonal family. In ProSpero, we approximate the intermediate importance weights using the *surrogate model*, which serves as a tractable proxy for the true likelihood. Moreover, our biologically constrained SMC restricts proposals to charge-compatible amino acids, making certain residues impossible to sample. In contrast, CloneBO does not enforce such constraints; although twisting reduces the probability of sampling undesirable residues, it does not eliminate them entirely.
>
> >"It would be of benefit if the authors situate their guidance method relative to other generative guidance methods [...]"
>
> Thank you for this helpful suggestion. We agree that situating our method in relation to existing guidance techniques would help readers better understand our contribution. In the revised version of the manuscript, we will include the following paragraph and expand it further with additional references and examples to strengthen its context and relevance.
>
> Classifier guidance is a technique used to steer generative models during inference, by incorporating gradients from an auxiliary classifier. The sampling is biased toward desired properties by adjusting the generative score with the gradient of the proxy classifier $\nabla_x \log p(y \mid x)$ [2]. However, this approach is not directly applicable in the discrete data domain, where gradients with respect to input tokens are not well-defined. Recently, Nisonoff et al. [3] proposed a technique allowing for classifier guidance in discrete diffusion and flow matching models. Discrete-domain guidance can also be addressed using SMC-based approaches, which guide sequence generation by maintaining a population of partial samples and resampling them based on their likelihood under a target distribution.
>
>
> > RAAs: In practice, what is the impact of using the RAA based on charge particles? [..]
>
> In our ablation study in Section 5.4, we already compare sampling from EvoDiff with and without RAAs constraints. For convenience, as well as given that both of those ablations are constant across noise levels, we summarized the results in the table below:
>
> |Method|Maximum fitness|Mean fitness|Diversity|Novelty|
> |-|-|-|-|-|
> |Evodiff|**0.570 ± 0.039**|**0.498 ± 0.024**|**7.83 ± 0.45**|**7.71 ± 1.18**|
> |Evodiff w/o RAA| 0.509 ± 0.012|0.356 ± 0.011|6.86 ± 0.50|4.11 ± 0.82|
>
> Motivated by your remark, we will extend our ablation by directly comparing the performance of ProSpero with and without RAAs constraints. The results are shown below:
>
> |SNR level|-25|-20|-15|-10|-5|0|
> |-|-|-|-|-|-|-|
> |ProSpero|**0.566 ± 0.030**|**0.586 ± 0.026**|**0.651 ± 0.032**|**0.679 ± 0.047**|**0.704 ± 0.016**|**0.706 ± 0.022**|
> |ProSpero w/o RAA|0.507 ± 0.025|0.555 ± 0.040|0.588 ± 0.029|0.605 ± 0.032|0.653 ± 0.042|0.666 ± 0.029|
>
> Both of these tables demonstrate the relevance of RAA constraints, particularly with respect to fitness and novelty. Thank you for prompting this valuable addition.
>
> > What value of n_sub was used in practice? [...]
>
> The number of masked sites in each benchmark was determined based on the sequence length, with the number sampled uniformly from the range of 5–15 residues for longer, and 3–10 for shorter sequences. The 5–15 residue masking range reflects common practice in rational design (e.g., for the GFP [4]) and was shown to perform well in silico by Kim et al. [5], who also explored masking 5% of residues. Based on this proportion, we reduced the range to 3–10 residues for shorter sequences like AAV, E4B, and Pab1. The reviewer’s intuition is correct, this parameter directly modulates novelty by controlling how far generated sequences deviate from the wild-type.
>
> > Line 147 defines $\pi^{(i)}$ as the order agnostic indices, why is it concatenated in that specific order? [...]
>
> The permutation order was chosen to prioritize charge classes with fewer valid options. The intuition behind this choice is that resolving the most constrained decisions early can help avoid suboptimal completions later in the sequence. We compared different orderings and summarized the results in the table below:
>
> |Ordering|AMIE|TEM|E4B|Pab1|AAV|GFP|UBE2I|LGK|
> |-|-|-|-|-|-|-|-|-|
> |Ascending |**0.246 ± 0.006**|1.231 ± 0.002|8.114 ± 0.037|**1.527 ± 0.254**|**0.720 ± 0.027**|**3.617 ± 0.003**|**2.993 ± 0.003**|**0.043 ± 0.002**|
> |Descending|0.244 ± 0.005|**1.232 ± 0.003**|8.139 ± 0.037|1.363 ± 0.141|0.706 ± 0.035|3.614 ± 0.003|2.991 ± 0.003|0.041 ± 0.003
> |Random|0.243 ± 0.005|**1.232 ± 0.002**|**8.164 ± 0.015**|1.338 ± 0.113|0.708 ± 0.029|3.614 ± 0.003|**2.993 ± 0.002**|0.042 ± 0.003|
>
> While the results suggest that the intuition is correct, the differences are modest, especially for longer sequences. Thank you for bringing attention to this aspect.
>
> Thank you for the thoughtful and constructive feedback, which helped us improve the manuscript. We hope our revisions address your concerns and would greatly appreciate it if you would consider **updating your evaluation of our submission**.
>
> #### References
> [1] A. N. Amin et al. Bayesian optimization of antibodies informed by a generative model of evolving sequences. ICLR 2025
>
> [2] P. Dhariwal and A. Nichol. Diffusion Models Beat GANs on Image Synthesis. NeurIPS 2021
>
> [3] H. Nisonoff. Unlocking Guidance for Discrete State-Space Diffusion and Flow Models. ICLR 2025
>
> [4] T. Hayes et al. Simulating 500 million years of evolution with a language model. Science 2025
>
> [5] H. Kim et al. Improved off-policy reinforcement learning in biological sequence design. 2024

---

> > ### Author Response · Authors · 2025-08-05
> >
> > Dear Reviewer,
> >
> > With the discussion period coming to an end, we wanted to ensure that our replies have satisfactorily resolved your questions and concerns. If there are any remaining areas that might need further clarification, please let us know. We sincerely value your time and thoughtful input.
> >
> > Respectfully,
> >
> > Authors

---

### Official Review · Reviewer_HtS2 · 2025-07-01

**Clarity:** 4
**Significance:** 2
**Originality:** 2
**Rating:** 2
**Confidence:** 4

**Summary:**

PROSPERO is method for black-box optimization of proteins that combines biological priors extracted from a pretrained evolutionary diffusion model (EvoDiff) and human knowledge (e.g. Restricted amino-acid classes based on biochemical properties), with a sampling approach that operates in a restricted exploration space observing the biologically plausible priors. The method is tested on a number of in silico benchmarks for black box biological sequence design and is shown to perform well on optimization and diversity metrics within these benchmarks against standard methods in the field.

**Questions:**

I suggest the authors develop the paper further by doing an in vitro experiment which can be done by an outside entity or collaborative lab (costs the same as 1000-10000 GPU hours), even for one empirical round.

Other than the in vitro validation, the main challenge/remedy or contribution would be to find protein optimization landscapes that show real distribution shift, and non-smooth optimization surfaces. Unfortunately at this time FLEXS and similar in silico landscapes are too low of a bar given what we know (helpful as a first check but not sufficient). The authors could think more critically about how to construct such landscapes, for instance using the Alphafold metrics and if they can make a convincing case about distribution shift in these landscapes, that's both a strong contribution and changes my view of the paper's promise.

**Ethical Concerns:**

["NO or VERY MINOR ethics concerns only"]

**Limitations:**

yes

**Paper Formatting Concerns:**

No concerns

**Quality:**

3

**Strengths And Weaknesses:**

Strength

The paper’s core thesis and application domain are relevant and correct. It should be helpful to incorporate biological priors into exploration algorithms, especially when one seeks to explore variants farther from the wild-type.

In terms of in silico benchmarking the work is well done and covers the important classes of algorithms. It also includes reasonable diversity metrics, which the method seems to perform well on.

The method also seems to be robust to noise, which is highly valuable.

The study execution within the in silico space of datasets, metrics, methods and evaluation is strong.

Weaknesses

My major concerns about the work is that positioning itself as a method for “far from wild type” design, makes it more necessary to show at least one empirical result. The standard has raised in the past 3-4 years on design algorithms to move beyond in silico optimization, and empirical validation is also relatively cheap and fast. See Gruver et al Protein Design with Guided Discrete Diffusion, NeurIPS 2023. It is very hard to evaluate methods that claim to improve OOD design without any real-world validation, which while correlated with in silico, does not show state of the art performance. While the approach is entirely plausible, whether the biological priors actually help in design is a question you’d only answer if you evaluate it with real biology. I think the improvement offered here (in optimization performance in silico) is insufficient to convince me it will be an advantage in the real world. For instance Damani et al Beyond the training set.. 2023 (https://arxiv.org/pdf/2311.05363), claim almost 100% of samples far from WT that score highly on a surrogate model similar to those are broken in real life. My experience is consistent with this view and I while I think the priors incorporated in this method may help this problem, there is no clear evidence it will.

More granularly, for instance, the AAV landscape from FLEXS that is used for evaluation is a synthetic landscape, not a real one. It is built from summing the effects of single residues (which were measured). This is a convex landscape with one peak that any hill climbing algorithm can eventually (if inefficiently solve). No real biology is this way. So showing that the algorithm can climb this hill is necessary but entirely insufficient in showing that it can design sequences far from the wild type. There is also no reason why the priors incorporated into the model should help in climbing this landscape, because it is only mildly connected to those biological rules.

As mentioned above the landscapes used here do not show strong distributional shifts.  Without directly addressing this point, the main claim of the paper on its ability to do optimization far from wild type is unconvincing.

Secondarily most protein design today occurs within 2-4 active learning rounds. Improved performance in that regime is far more valuable than performance improvements over more than 7 rounds of improvement

---

> ### Author Rebuttal · Authors · 2025-07-30
>
> We sincerely thank the reviewer for their thoughtful and detailed feedback.
>
> ## Regarding the primary concern about the lack of wet-lab validation
> We share the reviewer’s view on the importance of real-world validation, yet want to emphasize that experimental testing is beyond the scope of our current work. Here, we focus on establishing the methodological foundation of active learning-based, robust protein design, evaluating it on a wide variety of benchmarks. At the same time, we respectfully suggest that the absence of wet-lab results should not diminish the core methodological contribution of our work.  As the reviewer acknowledged, the paper’s central thesis and application domain are relevant, and our *in silico* benchmarking is strong and comprehensive.
>
> We thank the reviewer for highlighting the important and relevant work by Damani et al. [1]. We fully agree that great care must be taken when interpreting surrogate model predictions in regions far from the training distribution. That said, we believe the findings in Damani et al. [1] are not directly transferable to our setting. Their study and its empirical results focus on *offline* MBO. In contrast, our approach operates in an *online* active learning setting, where the surrogate is updated with oracle feedback. Crucially, it has been shown that active learning can significantly improve extrapolation and generalization in OOD regimes by enabling the model to iteratively incorporate data from previously unsupported areas of the search space [2]. We understand, however, that as long as the oracle is not a direct measure of biological function, concerns about over-optimism in unexplored regions remain justified.
>
> ## Regarding the landscapes, in particular AAV
>
> While indeed the AAV landscape is convex in the noise-free setting, it constitutes only one of the eight tasks we evaluate on. The remaining 7 landscapes are non-convex, rugged and more representative of the complexity of real biological functions. Moreover, in the noisy setting, the AAV landscape loses its convexity and resembles the Rough Mt. Fuji model, which has been shown to capture locally valid biological behavior around wild-type sequences in empirical studies [3]. We acknowledge the reviewer’s concern that success on this task alone may not be sufficient to support claims about generalization beyond wild-type neighborhoods. However, particularly in the noisy setting, it remains a valuable component of the evaluation suite.
>
>
> ## Regarding the insufficient distribution shifts on evaluated tasks
> We appreciate and acknowledge the reviewer's concern regarding the degree of distributional shift present in our evaluation tasks. To address this more directly, following helpful suggestions of the reviewer as well as Damani et al [1], we introduced another 3 tasks evaluating effectiveness of our method under various degrees of distribution shift.  In these experiments we used ESMFold as the oracle, which offers high-fidelity feedback better suited for performance assessment in more challenging and realistic settings.
>
> Specifically, from all UBE2I candidate sequences generated by different methods and scored with ESMFold, we selected those within a Hamming distance $\leq 5$ from the UBE2I wild-type to construct the initial dataset $D_0$, consisting of 2624 sequence---pTM score pairs. The average pTM score of $D_0=0.860 ± 0.029$. We then trained a surrogate model $f_\theta$ on $D_0$, and initiated the optimization not from the wild-type sequence, but from a sequence differing from the wild-type by a large number of 35 residues and of an initial pTM score of 0.70. Following the insightful suggestion of the reviewer, we run such optimization for only 4 iterations. The results are shown below:
>
> | Method | Maximum pTM|Mean pTM|Diversity| Novelty|
> |-|-|-|-|-|
> | AdaLead |0.796 ± 0.013|0.755 ± 0.011|8.83 ± 2.54| 8.36 ± 2.97|
> | PEX |0.807 ± 0.023|*0.760 ± 0.012*|6.14 ± 0.89| 4.45 ± 0.38|
> |GFN-AL-$\delta$CS |0.791 ± 0.010|0.729 ± 0.005| **16.92 ± 0.88**|9.56 ± 0.60|
> |MLDE|*0.810 ± 0.020*|0.752 ± 0.004|9.89 ± 1.11 |**20.88 ± 2.98**|
> | LatProtRL |0.787 ± 0.013|0.743 ± 0.003|6.32 ± 0.32| 5.90 ± 0.53|
> |ProSpero|**0.822 ± 0.027**|**0.777 ± 0.020**| *11.50 ± 1.62*|*17.74 ± 3.20*|
>
> where by novelty here we mean the distance of top 100 sequences to the starting point, not the original wild-type.
>
> Moreover, we additionally repeated the same experiment, but under an even more severe distribution shift, starting optimization from a sequence differing from the wild-type by a larger number of 75 residues and of an initial pTM score of 0.50. The results are shown below:
>
> | Method | Maximum pTM| Mean pTM| Diversity| Novelty|
> |-|-|-|-|-|
> |AdaLead | 0.593 ± 0.028| 0.526 ± 0.007| 14.26 ± 1.91| 7.66 ± 1.08|
> |PEX | 0.578 ± 0.014|0.518 ± 0.003|3.40 ± 0.07| 1.72 ± 0.04|
> |GFN-AL-$\delta$CS|0.630 ± 0.024|0.542 ± 0.006|**24.13 ± 1.47**|14.63 ± 1.16|
> |MLDE | *0.652 ± 0.059*| *0.572 ± 0.035*|13.10 ± 1.18| *21.68 ± 3.85*|
> |LatProtRL|0.560 ± 0.000|0.508 ± 0.003|2.24 ± 0.14| 1.78 ± 0.16|
> |ProSpero|**0.672 ± 0.031**|**0.599 ± 0.014**|*14.51 ± 1.99*|**22.03 ± 1.69**|
>
> Additionally, following the suggestion of Reviewer XJMJ, we repeated the experiment under the first distribution shift setting. To better reflect real-world constraints, we further limited the initial dataset by randomly selecting only 200 data points from $D_0$, resulting in a reduced initial training set with an average pTM score of $0.860±0.023$.
>
> |Method| Maximum pTM| Mean pTM| Diversity| Novelty|
> |-|-|-|-|-|
> |AdaLead|0.781 ± 0.016|0.742 ± 0.004|7.99 ± 1.39|5.95 ± 2.04|
> |PEX|*0.806 ± 0.013*|*0.752 ± 0.005*|6.77 ± 0.45|4.39 ± 0.51|
> |GFN-AL-$\delta$CS|0.782 ± 0.006|0.731 ± 0.006|**15.82 ± 1.77**|9.46 ± 1.69|
> |MLDE|0.782 ± 0.022|0.735 ± 0.025|9.39 ± 2.88|**16.97 ± 3.78**|
> |LatProtRL|0.792 ± 0.013|0.743 ± 0.001|6.25 ± 0.23| 5.57 ± 0.20|
> |ProSpero|**0.808 ± 0.017**|**0.763 ± 0.017**|*11.25 ± 2.51*|*15.87 ± 1.17*|
>
>
> These results demonstrate that our approach can robustly guide exploration toward high-fitness sequences, even under substantial distributional shifts, consistently outperforming competing methods. Moreover, in the setting of the more severe covariate shift with 75 mutations from the wild-type (which is roughly equivalent to half of the entire sequence), the performance gap between our approach and the majority of the baselines widens substantially.
>
> We trust that this additional evaluation directly addresses the reviewer’s concern about insufficient distributional shift in our original experiments. All three tables with the results for these novel landscapes with distribution shifts will be incorporated in the paper. We hope the reviewer will see this extended evidence and evaluation procedure as a meaningful contribution and reconsider their assessment of the paper.
>
>
> ## Regarding the performance within early active learning rounds
>
> We thank the reviewer for raising this important and practically relevant point about the typical number of active learning rounds used in real-world protein design workflows. Due to space constraints, we present only the most relevant metric below, namely the maximum fitness achieved within 4 iterations. The revised version of the manuscript will include the full set of results limited to 2 to 4 iterations across all benchmarks.
>
> | Method|AMIE|TEM|E4B| Pab1|AAV|GFP|UBE2I|LGK|
> |-|-|-|-|-|-|-|-|-|
> | AdaLead    | 0.232 ± 0.003 | 1.227 ± 0.004 | 7.962 ± 0.071 | **1.397 ± 0.329** | 0.596 ± 0.014 | 3.580 ± 0.003 | 2.982 ± 0.002 | *0.032 ± 0.002* |
> | PEX        | **0.242 ± 0.001** | **1.231** ± 0.001 | *7.971 ± 0.078* | 1.064 ± 0.071 | *0.604 ± 0.018* | *3.597 ± 0.002* | *2.987 ± 0.002* | 0.030 ± 0.001 |
> | GFN-AL-δCS | 0.160 ± 0.048 | 0.563 ± 0.119 | 7.859 ± 0.047 | 1.035 ± 0.094 | 0.596 ± 0.010 | 3.584 ± 0.005 | 2.972 ± 0.013 | 0.032 ± 0.002 |
> | MLDE       | 0.231 ± 0.006 | *1.229 ± 0.000* | 7.821 ± 0.063 | 0.866 ± 0.024 | 0.555 ± 0.000 | 3.589 ± 0.003 | 2.978 ± 0.000 | 0.028 ± 0.003 |
> | LatProtRL  | 0.224 ± 0.000 | *1.229 ± 0.000* | 7.751 ± 0.016 | 1.031 ± 0.129 | 0.565 ± 0.011 | 3.589 ± 0.003 | 2.982 ± 0.001 | 0.020 ± 0.000 |
> | ProSpero   | *0.236 ± 0.007* | *1.229 ± 0.001* | **7.978 ± 0.055** | *1.202 ± 0.129* | **0.635 ± 0.019** | **3.602 ± 0.002** | **2.989 ± 0.002** | **0.036 ± 0.001** |
>
> Notably, even when restricted to a smaller number of iterations, our approach consistently ranks first or second across all tasks, highlighting its effectiveness in low-iteration regimes commonly encountered in practice, as rightly pointed out by the reviewer.
>
>
> We once again sincerely thank the reviewer for their thoughtful and constructive feedback, which helped us identify and address a key shortcoming in the original submission. In particular, the suggestion to better evaluate performance under distributional shift motivated the introduction of three new tasks designed to test robustness in more realistic and challenging OOD settings. We believe these additions significantly strengthen the paper and directly respond to the concerns raised. We hope the reviewer finds that these revisions adequately address their main points, and we would be grateful if they would consider **updating their evaluation of the work in light of these improvements**.
>
> #### References
> [1] F. Damani et al. Beyond the training set: an intuitive method for detecting distribution shift in model-based optimization. 2023.
>
> [2] E. R. Antoniuk et al. Active Learning Enables Extrapolation in Molecular Generative Models. 2025.
>
> [3] S. Sinai et al. AdaLead: A simple and robust adaptive greedy search algorithm for sequence design. 2020.

---

> > ### Author Response · Authors · 2025-08-05
> >
> > Dear Reviewer,
> >
> > With the discussion period concluding, we would like to confirm whether our responses and the extended set of experiments have addressed your concerns. We hope you will find these additions to meaningfully strengthen the paper. If any questions remain or further clarification is needed, please let us know. We deeply value your time and detailed feedback.
> >
> > Respectfully,
> >
> > Authors

---

> > ### Comment · Reviewer_HtS2 · 2025-08-06
> >
> > I appreciate the author's effort in addressing my secondary concerns.
> >
> > > That said, we believe the findings in Damani et al. [1] are not directly transferable to our setting. Their study and its empirical results focus on offline MBO.
> >
> > In molecular design, all "online" MBO is a series of "offline" MBO with a series of measurements in between. When you have more rounds (and most real experiments are 2-3 rounds, so not much room for exploration), you can be more conservative in surrogate exploration and more information-seeking in the ground truth space to start out and over time become more explorative in surrogate space and more optimizing in ground truth space. My point is, it is unclear to me if your method will fail at the very first round. ESMFold's landscape (like all surrogate landscapes known to me) is very smooth compared to any real fitness space.
> >
> > As a person that has implemented and tested many of these method in the real world, unfortunately, I can not accept any paper that claims to improve on the (current) in silico benchmarks alone.

---

> > > ### Author Response · Authors · 2025-08-07
> > >
> > > We thank the reviewer for their continued engagement in the discussion of our paper.
> > >
> > > > “In molecular design, all "online" MBO is a series of "offline" MBO with a series of measurements in between [...]”
> > >
> > > We agree with the reviewer that, in practice, online MBO can be viewed as a sequence of offline steps interleaved with measurements. However, we maintain that the specific observations made in Damani et al. [1] are not directly transferable to our setting. In their work, the optimization operates in a single round, and as reviewer noticed, as such they cannot allow themselves for more conservative surrogate exploration. As a consequence, their designed sequences exhibit a mode edit distance of 12 from the wild-type, which corresponds to roughly 20% of the total sequence length (63 residues). In our setting, the average per-round edit distance remains closer to 5%. We emphasize that the failure to generate functional sequences in Damani et al. arises in part from the particularly high distribution shifts imposed on the surrogate due to the single-step setting. For this reason, we believe that statements such as
> > >
> > > “Damani et al. [...] claim almost 100% of samples far from WT that score highly on a surrogate model similar to those are broken in real life”
> > >
> > > should be interpreted with care when extrapolated to our setting, as they reflect **a specific experimental regime that differs meaningfully from our setup in both design constraints and exploration dynamics**.
> > >
> > > > I can not accept any paper that claims to improve on the (current) in silico benchmarks alone.
> > >
> > > As previously suggested by the reviewer's feedback, with ESMFold we constructed a novel benchmark with a strong distribution shift. The results clearly indicate that ProSpero is robust to such strong shifts and can operate in OOD settings. We are also convinced that large foundation models such as ESMfold, are as good as it gets for providing reliable feedback in an in silico setting.
> > >
> > > While we ourselves are practitioners and are strongly convinced of experimental testing as the ultimate showcase of the methods' strengths, we are convinced that **blocking publications with in silico validation on machine learning venues would be detrimental to the field, as it will limit the development driven by AI labs.**
> > >
> > > Taken together, we are highly grateful for the reviewer's suggestion to develop a dedicated distribution shift benchmark and demonstrate the advantages of our approach on this benchmark. We are also much more optimistic than the reviewer regarding the adequacy of in silico validation, which is strongly backed up by our previous positive experience, where strong in silico results translated into successful experimental outcomes.
> > >
> > > #### References
> > >
> > > [1] F. Damani et al. Beyond the training set: an intuitive method for detecting distribution shift in model-based optimization. 2023.

---

### Official Review · Reviewer_gUKh · 2025-07-03

**Clarity:** 2
**Significance:** 3
**Originality:** 3
**Rating:** 4
**Confidence:** 2

**Summary:**

The paper presents PROSPERO, an active learning framework for protein sequence design that aims to generate sequences with high fitness, novelty, and biological plausibility. Its key innovations are: (1) iterative sequence design as inference-time guidance of a pre-trained generative model within an active learning loop that incorporates biological priors; (2) a targeted masking strategy that edits fitness-relevant residues while protecting critical sites; and (3) biologically constrained Sequential Monte Carlo (SMC) sampling, which restricts mutations based on explicit priors to improve discovery of high-fitness, novel sequences even when the surrogate model is imperfect. Experiments show that PROSPERO consistently outperforms or matches existing methods across diverse tasks, and is robust to surrogate model misspecification.

**Questions:**

refer to above

**Ethical Concerns:**

["NO or VERY MINOR ethics concerns only"]

**Final Justification:**

a sound paper for protein sequence design with relatively rough writing.

**Limitations:**

yes

**Quality:**

3

**Strengths And Weaknesses:**

Strengths

1. The PROSPERO framework is clearly presented, with comprehensive technical details and good structure. Key components—such as targeted masking based on alanine scanning and biologically-constrained SMC sampling—are introduced in detail to boost the reproducibility.

2. The paper conducts a comprehensive experimental evaluation including eight diverse protein engineering tasks, comparing PROSPERO to ten strong baselines. The ablation is comprehensive, and the results consistently demonstrate the effectiveness of  the proposed approach with an emphasis on exploration.

3. By guiding a frozen generative model with an active learning-updated surrogate and introducing targeted, biologically-aware masking, PROSPERO advances protein design. The approach effectively addresses the trade-off between novelty and biological plausibility, with broad potential impact on protein engineering applications.

Weaknesses

1.  Some technical components—such as SMC unrolling and perplexity correction—are formally described but may lack intuitive explanations for readers less familiar with these methods. The comparison with all baselines is somewhat uneven, limiting contextual understanding.

2.  While innovative in integrating various ideas, components like SMC and charge-based grouping are adapted from prior work, and experimental scope is limited to controlled scenarios. PROSPERO’s performance on more complex or out-of-distribution tasks remains untested.

---

> ### Author Rebuttal · Authors · 2025-07-30
>
> We thank the reviewer for their positive and thoughtful feedback.
>
> > Some technical components [...] are formally described but may lack intuitive explanations for readers less familiar with these methods.
>
>
> While the full details of the SMC procedure are included in the Appendix C, we agree that a more intuitive explanation in the main text could benefit readers less familiar with the method. We will include the following changes in the revised version of our work:
> - In Equation 1 of Section 4.2, we will revise the notation of the target distribution from $\gamma(x)$ to $\gamma(x_{1:L})$ , and update all other occurrences of $x$ accordingly within the introductory part of Section 4.2. This change makes explicit that the target is defined over a sequence of residues, improving clarity and alignment with the sequential structure of SMC
> - We will begin the L144-167 part with the following paragraph: Rather than trying to directly sample from complex, high-dimensional target distribution $\gamma(x_{1:L})$, in ProSpero we perform approximate inference using SMC, which decomposes the sampling process into a sequence of simpler, unnormalized, intermediate target distributions $\{\tilde{\gamma}(x_{1:l})\}_{l=1}^{L}$ (for background see Appendix C). This allows us to progressively build up sequences residue by residue, while maintaining a tractable approximation of the full design objective"
> - We will conclude the line 162 as follows: “Finally, the unnormalized importance weights … , with the perplexity correction term penalizing sequences that are improbable under the true prior, correcting bias introduced by the constrained sampling”
>
> We hope this revision will make the section clearer and more accessible to the readers.
> >  The comparison with all baselines is somewhat uneven, limiting contextual understanding.
>
> We would like to emphasize that  our evaluation covers eight diverse protein engineering tasks and in a fair manner compares against ten baselines spanning a wide range of algorithmic approaches, including evolutionary strategies, Bayesian optimization, reinforcement learning, GFlowNets, probabilistic models, and machine-learning-guided directed evolution. In addition to fitness optimization, we show tradeoffs to  exploration capabilities, biological plausibility and robustness to surrogate noise, resulting in a comprehensive and multi-faceted comparison. We believe this breadth offers a strong contextual understanding of the advantages of ProSpero relative to existing methods.
>
> > While innovative in integrating various ideas, components like SMC and charge-based grouping are adapted from prior work [...]
>
> We agree that SMC has been applied to protein design in prior works [1, 2]. We would like to clarify, however, that introduced by us biologically-constrained SMC represents a novel and principled extension: it allows biological priors to be incorporated directly into the proposal distributions. To specifically emphasize the impact of this contribution, we will complement our existing ablation study with a direct comparison between our biologically-constrained SMC and an unconstrained formulation:
> | SNR level    | -25       | -20       | -15       | -10       | -5       | 0       |
> |-----------|--------------|--------------|--------------|--------------|--------------|--------------|
> | ProSpero | **0.566 ± 0.030** | **0.586 ± 0.026** | **0.651 ± 0.032** | **0.679 ± 0.047** | **0.704 ± 0.016** | **0.706 ± 0.022** |
> | ProSpero w/o RAA | 0.507 ± 0.025 | 0.555 ± 0.040 | 0.588 ± 0.029 | 0.605 ± 0.032 | 0.653 ± 0.042 | 0.666 ± 0.029 |
>
> Additionally, we note that reduced amino acid alphabets (RAAs) are by no means standard or widely adopted in this context. Specifically, we refer to the work of Liang et al. [3], concluded with "[...] RAA alphabets have not been fully and maturely used in current research." Given this, we believe that our biologically informed use of RAAs represents an impactful contribution in its own right, representing a notable use case.
>
> > PROSPERO’s performance on more complex or out-of-distribution tasks remains untested.
>
> Thank you for raising this relevant point. To address your and other reviewers concerns, we introduced another 3 tasks evaluating effectiveness of our method under various degrees of distribution shift. In these experiments we used ESMFold as the oracle, which offers high-fidelity feedback better suited for performance assessment in more challenging and realistic settings.
>
> Specifically, from all UBE2I candidate sequences generated by different methods and scored with ESMFold, we selected those within a Hamming distance $\leq 5$ from the UBE2I wild-type to construct the initial dataset $D_0$, consisting of 2624 sequence---pTM score pairs. The average pTM score of $D_0=0.860 ± 0.029$. We then trained a surrogate model $f_\theta$ on $D_0$, and initiated the optimization not from the wild-type sequence, but from a sequence differing from the wild-type by a large number of 35 residues and of an initial pTM score of 0.70. We then run such set optimization for 4 iterations. The results are shown below:
>
> | Method | Maximum pTM|Mean pTM|Diversity| Novelty|
> |-|-|-|-|-|
> | AdaLead |0.796 ± 0.013|0.755 ± 0.011|8.83 ± 2.54| 8.36 ± 2.97|
> | PEX |0.807 ± 0.023|*0.760 ± 0.012*|6.14 ± 0.89| 4.45 ± 0.38|
> |GFN-AL-$\delta$CS |0.791 ± 0.010|0.729 ± 0.005| **16.92 ± 0.88**|9.56 ± 0.60|
> |MLDE|*0.810 ± 0.020*|0.752 ± 0.004|9.89 ± 1.11 |**20.88 ± 2.98**|
> | LatProtRL |0.787 ± 0.013|0.743 ± 0.003|6.32 ± 0.32| 5.90 ± 0.53|
> |ProSpero|**0.822 ± 0.027**|**0.777 ± 0.020**| *11.50 ± 1.62*|*17.74 ± 3.20*|
>
> where by novelty here we mean the distance of top 100 sequences to the starting point, not the original wild-type.
>
> Moreover, we additionally repeated the same experiment, but under an even more severe distribution shift, starting optimization from a sequence differing from the wild-type by a larger number of 75 residues and of an initial pTM score of 0.50. The results are shown below:
>
> | Method | Maximum pTM| Mean pTM| Diversity| Novelty|
> |-|-|-|-|-|
> |AdaLead | 0.593 ± 0.028| 0.526 ± 0.007| 14.26 ± 1.91| 7.66 ± 1.08|
> |PEX | 0.578 ± 0.014|0.518 ± 0.003|3.40 ± 0.07| 1.72 ± 0.04|
> |GFN-AL-$\delta$CS|0.630 ± 0.024|0.542 ± 0.006|**24.13 ± 1.47**|14.63 ± 1.16|
> |MLDE | *0.652 ± 0.059*| *0.572 ± 0.035*|13.10 ± 1.18| *21.68 ± 3.85*|
> |LatProtRL|0.560 ± 0.000|0.508 ± 0.003|2.24 ± 0.14| 1.78 ± 0.16|
> |ProSpero|**0.672 ± 0.031**|**0.599 ± 0.014**|*14.51 ± 1.99*|**22.03 ± 1.69**|
>
> Additionally, following the suggestion of Reviewer XJMJ, we repeated the experiment under the first distribution shift setting. To better reflect real-world constraints, we further limited the initial dataset by randomly selecting only 200 data points from $D_0$, resulting in a reduced initial training set with an average pTM score of $0.860±0.023$.
>
> |Method| Maximum pTM| Mean pTM| Diversity| Novelty|
> |-|-|-|-|-|
> |AdaLead|0.781 ± 0.016|0.742 ± 0.004|7.99 ± 1.39|5.95 ± 2.04|
> |PEX|*0.806 ± 0.013*|*0.752 ± 0.005*|6.77 ± 0.45|4.39 ± 0.51|
> |GFN-AL-$\delta$CS|0.782 ± 0.006|0.731 ± 0.006|**15.82 ± 1.77**|9.46 ± 1.69|
> |MLDE|0.782 ± 0.022|0.735 ± 0.025|9.39 ± 2.88|**16.97 ± 3.78**|
> |LatProtRL|0.792 ± 0.013|0.743 ± 0.001|6.25 ± 0.23| 5.57 ± 0.20|
> |ProSpero|**0.808 ± 0.017**|**0.763 ± 0.017**|*11.25 ± 2.51*|*15.87 ± 1.17*|
>
>
> These results demonstrate that our approach can robustly guide exploration toward high-fitness sequences, even under substantial distributional shifts, consistently outperforming competing methods. Moreover, in the setting of the more severe covariate shift with 75 mutations from the wild-type (which is roughly equivalent to half of the entire sequence), the performance gap between our approach and the majority of the baselines widens substantially.
>
> We trust that this additional evaluation directly addresses the reviewer’s concern about insufficient distributional shift in our original experiments. All three tables with the results for these novel landscapes with distribution shifts will be incorporated in the paper.
>
>
> Thank you for your constructive and meaningful feedback, which has helped strengthen the manuscript. We hope our responses have addressed your concerns, and would greatly appreciate it if you would consider **raising your rating of our submission** accordingly.
>
> #### References
> [1] A. N. Amin et al. Bayesian optimization of antibodies informed by a generative model of evolving sequences. ICLR 2025.
>
> [2] L. Wu et al. Practical and Asymptotically Exact Conditional Sampling in Diffusion Models. NeurIPS 2023.
>
> [3] Y. Liang et al. Research Progress of Reduced Amino Acid Alphabets in Protein Analysis and Prediction. Computational and Structural Biotechnology Journal, 2022.

---

> > ### Author Response · Authors · 2025-08-05
> >
> > Dear Reviewer,
> >
> > As we approach the end of the discussion period, we wish to confirm that our responses have addressed your feedback in full. If you feel any issues remain or would like additional clarification on certain points, we would be happy to provide it. Thank you for your time and careful review.
> >
> > Respectfully,
> >
> > Authors

---

> > ### Comment · Reviewer_gUKh · 2025-08-08
> > **Response to Rebuttal**
> >
> > Thanks for the detailed rebuttal and additional information.
> > The additional results and discussions make the background and empirical contribution clear to me. I will keep my positive rating.

---

> > > ### Author Response · Authors · 2025-08-08
> > >
> > > We are glad that our additional results and discussion helped clarify our contributions. Thank you for maintaining your positive assessment of our work.

---

### Official Review · Reviewer_XJMJ · 2025-07-11

**Clarity:** 3
**Significance:** 3
**Originality:** 3
**Rating:** 5
**Confidence:** 3

**Summary:**

In this work a new active learning framework is proposed for the design of protein sequences. The proposed method explores functionally important residue sites instead of random sites, which are identified inspired by a technique used in the lab to perform mutagenesis. Additionally, a pre-trained generative model along with a surrogate function of the property is used to sample sequences having optimized properties, while maintaining charge-class similarity to the wild-type at functionally important residue sites. This allows for the generation of biologically plausible sequences in regions of the design space where surrogate predictions may be unreliable.

**Questions:**

1) How many sites are masked in each benchmark and how is this number determined?

2) In each active learning round, only the best sample/sequence is used for masking and proposal of new candidates. Isn't this limiting the exploration capability of the model? I understand that simplifying the algorithm by sticking to the best sequence is a good design if it is working, but have you performed any ablation study on the number of top sequences used for candidate generation?

3) I find the description for "Sampling from ... using Sequential Monte Carlo", lines 144-167 pretty hard to follow. It would be nice to explain what the overall procedure should look like and why before getting into its steps.

4) Is the extension beyond wild-type neighborhood only measured by the novelty metric (average hamming distance between top 100 candidates and a starting sequence (wild-type))? What is the average hamming distance between the starting wild-type sequence and the rest of the sequences in the initial dataset? Or what is the average distance between the top 100 and the initial dataset?
The reason I am asking about the distance from the initial dataset is because it is discussed that the model is robust to the surrogate misspecification/uncertainty, so I would like to know if there is a way to quantify the extent of exploration beyond surrogate comfort zone (initial dataset) for the benchmark in Table 1.

5) If I understood it correctly, all experiments are performed with fixed initial datasets. So the variability in the performance only comes from the model stochasticity. Have you examined how the initial dataset affects the model performance? This matters when the size of the initial dataset is small.
Your algorithm starts by taking the best sample from the initial dataset, so having a smaller initial dataset, with less coverage of the landscape, could make the optimization harder. Based on the information included in the Appendix, the initial datasets are quite large (3K, 5K, 6K, 7K, 10K, 15K) which is NOT the case in many practical settings. Have you examined the variability of your model's performance on smaller initial datasets (<< 1000, e.g., 200)?
The size of the initial dataset affects the quality of the trained surrogate model as well as the search in the design space depending on how the initial samples are distributed within the landscape, as discussed in prior work [1]. I do not expect you to add extra experiments on this but how do you expect your model to behave for small initial datasets?

6) I liked the idea and reasoning for scoring residues by Alanine replacement. Did you try other amino acids as baselines? I understand that this is not backed by what experimentalists would do but computationally, one would like to know how significant the choice of Alanine is over the rest of the amino acids.

7) Nice-to-haves: Missed citation for a recent generative-based approach [1] developed to address  practical challenges of protein design.

References:

[1] Ghaffari et al. Robust Model-Based Optimization for Challenging Fitness Landscapes. ICLR 2024.
https://arxiv.org/pdf/2305.13650.

**Ethical Concerns:**

["NO or VERY MINOR ethics concerns only"]

**Final Justification:**

All my questions were addressed.

**Quality:**

3

**Strengths And Weaknesses:**

What I really liked about this work is basing the model design on how experimentalist scan for important sites in mutagenesis experiments.

---

> ### Author Rebuttal · Authors · 2025-07-30
>
> Thank you for taking the time to provide such a detailed and insightful review of our manuscript.
>
> ## Q1: Number of mutational sites
> The number of masked sites in each benchmark was determined based on the sequence length, with the number sampled uniformly from the range of 5–15 residues for longer, and 3–10 for shorter sequences. The range of 5–15 residues aligns with ranges commonly used by experimentalists in rational design for example on the GFP task [1], making this another aspect of our framework that reflects mutagenesis-inspired behavior. Kim et al. [2] have demonstrated that this masking range performs well *in silico*. Additionally, they experiment with masking 5% of the total residues in a sequence. Based on this, for shorter sequences, such as AAV, E4B, and Pab1, we reduced the masking range to 3–10 amino acids.
>
>
> ## Q2: Ablation study on the number of top starting sequences for the candidate generation
> Thank you for the helpful suggestion. We added an ablation on LGK, the task requiring generation of the longest sequences, to test the effect of varying the number of top starting sequences (1–128). Results are shown below:
>
> |N starting sequences| Maximum fitness| Mean fitness | Diversity     | Novelty      |
> |-|-|-|-|-|
> |1| **0.043 ± 0.002**   | **0.040 ± 0.002**   | 17.25 ± 5.64  | **74.33 ± 7.75** |
> |4| 0.041 ± 0.001   | 0.039 ± 0.001   | 19.30 ± 4.58  | 70.20 ± 6.05 |
> |16| 0.038 ± 0.002   | 0.037 ± 0.002   | 19.09 ± 5.80  | 64.86 ± 5.44 |
> |64| 0.037 ± 0.002   | 0.034 ± 0.001   | 29.13 ± 6.97  | 56.38 ± 7.06 |
> |128| 0.033 ± 0.002   | 0.030 ± 0.002   | **33.89 ± 9.54**  | 50.45 ± 5.52 |
>
> Fewer starting points drive deeper, more directed exploration, resulting in higher novelty and fitness but lower diversity. In contrast, more starting points promote broader, more diffuse exploration, increasing diversity but limiting how far any single trajectory moves from the wild-type across subsequent optimization rounds.
>
> As suspected by the reviewer, this new table brings important insights into the tradeoffs associated with the number of initial sequences and the exploration capabilities and we will incorporate it into the final version of the paper.
>
>
> ## Q3: Clarity of lines 144-167
> Thank you for the comment. We understand that this section may be difficult to follow without additional context. To support readers unfamiliar with Sequential Monte Carlo, we already refer to Appendix C in line 146, which provides a more detailed mathematical background and explains the overall procedure. To further improve the clarity of this section, the revised version of our work will feature the following changes:
> - In Equation 1 of Section 4.2, we will revise the notation of the target distribution from $\gamma(x)$ to $\gamma(x_{1:L})$ , and update all other occurrences of $x$ accordingly within the introductory part of Section 4.2. This change makes explicit that the target is defined over a sequence of residues, improving clarity and alignment with the sequential structure of SMC
> - We will begin the L144-167 part with the following paragraph: Rather than trying to directly sample from complex, high-dimensional target distribution $\gamma(x_{1:L})$, in ProSpero we perform approximate inference using SMC, which decomposes the sampling process into a sequence of simpler, unnormalized, intermediate target distributions $\{\tilde{\gamma}(x_{1:l})\}_{l=1}^{L}$ (for background see Appendix C). This allows us to progressively build up sequences residue by residue, while maintaining a tractable approximation of the full design objective"
> - We will conclude the line 162 as follows: “Finally, the unnormalized importance weights […] , with the perplexity correction term penalizing sequences that are improbable under the true prior, correcting bias introduced by the constrained sampling”
>
> We hope this revision will make the section clearer and more accessible to the readers.
>
>
>
> ## Q4: Other measures of exploration
> Thank you for this insightful suggestion. We agree that the inclusion of the proposed Hamming distances will help to quantify the extent of exploration of our approach. We have computed the average Hamming distance between each starting sequence and its corresponding initial dataset.
>
> || AMIE | TEM  | E4B  | Pab1 | AAV  | GFP   | UBE2I | LGK  |
> |-|--|--|--|--|--|--|--|--|
> | WT distance to initial dataset | 2.00 | 2.00 | 5.42 | 3.95 | 5.05 | 42.87 | 2.00 | 2.00 |
>
> As shown, distances for the majority of the tasks are small. This is expected given that most datasets were created by wet-lab scientists applying a limited number of mutations to the wild-type sequence and measuring the activity of such mutated variants. A notable exception is the GFP task, where the initial dataset was created *in silico* by Kim et al. [2] through random mutagenesis and scoring via the oracle.
>
> We have also computed the average distances between top 100 sequences and the initial datasets:
> | Method| AMIE| TEM | E4B|Pab1| AAV| GFP| UBE2I| LGK|
> |-|-|-|-|-|-|-|-|-|
> | AdaLead| 5.45 ± 0.62  | 3.04 ± 0.03  | 9.21 ± 0.58  | 11.84 ± 1.35 | *14.09 ± 1.43* | 70.14 ± 4.74 | 4.83 ± 0.48  | 16.31 ± 7.22 |
> | PEX| 7.18 ± 1.08  | 3.78 ± 0.21  | 9.24 ± 0.52  | 9.06 ± 0.71  | 11.75 ± 1.27 | 47.89 ± 1.09 | 6.51 ± 0.82  | 10.43 ± 1.38 |
> | GFN-AL-$\delta$CS | 8.47 ± 0.40  | **10.19 ± 0.67** | 10.25 ± 1.55 | 12.88 ± 1.27 | 13.93 ± 0.32 | *70.98 ± 3.02* | 8.99 ± 2.97  | *63.46 ± 4.33* |
> | MLDE| *20.83 ± 2.78* | *5.99 ± 0.42*  | **16.91 ± 3.39** | *13.12 ± 3.04* | 9.14 ± 0.23  | 62.02 ± 4.40 | *11.53 ± 2.56* | 51.91 ± 7.99 |
> | LatProtRL| 2.10 ± 0.04  | 2.01 ± 0.01  | 7.59 ± 0.46  | 6.62 ± 0.84  | 7.84 ± 0.36  | 53.02 ± 1.78 | 2.51 ± 0.11  | 2.00 ± 0.00  |
> | ProSpero| **22.92 ± 3.30** | 5.31 ± 0.53  | *13.99 ± 0.95* | **15.44 ± 3.42** | **18.88 ± 1.47** | **75.09 ± 3.10** | **18.33 ± 5.08** | **75.95 ± 7.84** |
>
> Such evaluated performance closely resembles the one in Table 3, with ProSpero achieving larger distances than compared models for the majority of the tasks. These results underscore the capabilities of our approach to explore regions of the search space beyond the initial training data, even in cases such as the GFP task, where the initial sequences are not as densely clustered around the wild-type. This new table will be featured in the revised paper.
>
>
> ## Q5: Initial dataset size influence on the performance
> Thank you for raising this important point. We agree that small initial datasets are common in practice. Following your suggestion, as well as suggestions of other reviewers, we've added an additional task that better reflects real-world settings.
>
> Specifically, from all UBE2I candidate sequences generated by different methods and scored with ESMFold throughout our experiments, we randomly selected *200* of those within a Hamming distance $\leq 5$ from the UBE2I wild-type to construct the initial dataset $D_0$, where the average pTM score of $D_0 = 0.860 ± 0.023$. We then trained a surrogate model $f_\theta$ on $D_0$, and initiated the optimization not from the wild-type sequence, but from a sequence differing from the wild-type by a large number of 35 residues and of an initial pTM score of 0.70. We run such optimization for only 4 iterations, employing ESMFold as an oracle. The results are shown below:
>
> |Method| Maximum pTM| Mean pTM| Diversity| Novelty|
> |-|-|-|-|-|
> |AdaLead| 0.781 ± 0.016   | 0.742 ± 0.004   | 7.99 ± 1.39   | 5.95 ± 2.04  |
> |PEX | *0.806 ± 0.013*   | *0.752 ± 0.005*   | 6.77 ± 0.45   | 4.39 ± 0.51  |
> |GFN-AL-$\delta$CS| 0.782 ± 0.006   | 0.731 ± 0.006   | **15.82 ± 1.77**  | 9.46 ± 1.69  |
> |MLDE| 0.782 ± 0.022   | 0.735 ± 0.025   | 9.39 ± 2.88   | **16.97 ± 3.78** |
> |LatProtRL | 0.792 ± 0.013   | 0.743 ± 0.001   | 6.25 ± 0.23   | 5.57 ± 0.20  |
> |ProSpero | **0.808 ± 0.017**   | **0.763 ± 0.017**   | *11.25 ± 2.51*  | *15.87 ± 1.17*  |
>
> We believe the strong performance of our approach in comparison to the baselines is due to the incorporation of biological priors, which enable efficient exploration even when the surrogate model is weak, whether because of limited training data or distribution shifts. We will include this result in a revised version of the paper.
>
> ## Q6: Amino acid choice for replacement-based residue scoring
> In our work, we confirmed the advantage of residue scoring using Alanine scan over random amino acid substitutions, as shown in our ablation studies in Section 5.4.
> For convenience, we summarized the key results in the table below:
>
> |SNR level|-25|-20| -15| -10| -5| 0|
> |-|-|-|-|-|-|-|
> | ProSpero   | **0.566 ± 0.03**  | **0.586 ± 0.026** | **0.651 ± 0.032** | **0.679 ± 0.047** | **0.704 ± 0.016** | **0.706 ± 0.022** |
> | ProSpero w/o TM     | 0.554 ± 0.021 | 0.581 ± 0.028 | 0.597 ± 0.027 | 0.672 ± 0.021 | 0.677 ± 0.030 | 0.649 ± 0.028 |
>
> We did not perform systematic one-to-one comparisons with other specific amino acids. This is because alanine scanning is a well-established strategy in experimental mutagenesis, whereas there is little principled justification for selecting any other single amino acid as a universal substitute with neutral properties. As such, alternative choices lack clear biological or methodological motivation, making direct comparisons difficult to interpret meaningfully.
>
>
> ## Q7: Missing citation
>
> Thank you for the suggestion. We agree the cited work is relevant and will include it in the revised manuscript.
>
> Thank you for the thoughtful and constructive comments, which greatly helped us improve the manuscript. We hope we have addressed your concerns, and if so, we would be grateful if you would consider **raising your rating of our submission**.
>
> #### References
> [1] T. Hayes et al. Simulating 500 million years of evolution with a language model. Science 2025.
>
> [2] H. Kim et al. Improved off-policy reinforcement learning in biological sequence design. 2024.

---

> > ### Author Response · Authors · 2025-08-05
> >
> > Dear Reviewer,
> >
> > As the discussion period comes to an end, we wanted to confirm whether our responses have sufficiently addressed your questions. If you have any remaining concerns or if there are points that could benefit from further clarification, please let us know. We greatly value your time and thoughtful feedback.
> >
> > Respectfully,
> >
> > Authors

---

> > ### Comment · Reviewer_XJMJ · 2025-08-08
> > **Response to Rebuttal**
> >
> > Thanks for the answers provided. I will raise my score to 5.

---

> > > ### Author Response · Authors · 2025-08-08
> > >
> > > We are glad our responses addressed your questions, and we sincerely thank you for your support of our work.

---

### Comment · Area_Chair_uH41 · 2025-08-06
**Participation in the rebuttal**

Dear reviewers,

Please engage in the discussion with the authors. The discussion period will end in a few days.

Best,
AC

---

### Note · Authors · 2025-08-13

We are sincerely grateful to the area chair for overseeing the review process and to all the reviewers for their feedback and engagement throughout the discussion period.

**Summary of Rebuttal Contributions**:
- Introduced benchmarks with strong distribution shifts, where ProSpero consistently outperformed state-of-the-art methods
- Conducted additional ablations, quantifying the effect of varying the number of top starting sequences on exploration–exploitation trade-offs and showcasing the benefits of targeted masking over random masking
- Demonstrated competitive or superior results of ProSpero within 4 iterations, matching typical constraints in protein design campaigns
- Refined the presentation of the SMC formulation and notation, clearly situating our approach relative to prior work
- Added metrics capturing the extent of exploration achieved by ProSpero beyond wild-type neighborhoods

**Reviewer Evaluations**:
- XJMJ: Appreciated the mutagenesis-inspired design of targeted masking and raised their score to 5 following the rebuttal
- gUKh: Praised our comprehensive evaluation, noting that the “*... results consistently demonstrate the effectiveness of the proposed approach …*” and that it has “*... broad potential impact on protein engineering applications.*”. Maintained a positive score
- HtS2: Recognized the strength and breadth of our *in silico* evaluation, remained concerned about the lack of wet-lab validation
- Dp5Q: Noted strong results relative to benchmarks, thorough evaluation of generated sequences for biological validity, and high reproducibility with open-source code

**Significance of this work**:

ProSpero addresses the key challenge in AI-driven protein design of enabling exploration beyond wild-type neighborhoods. It achieves this by incorporating biological priors through:
- **Inference-time guidance of a pre-trained pLM with a surrogate updated in an active learning loop**, enabling seamless integration of biological priors into online optimization
- **Targeted masking** of fitness-relevant residues while preserving key structural sites, avoiding disruptions caused by uninformed edits
- **Biologically-constrained SMC sampling** that restricts proposals to wild-type-like residues, improving the likelihood of finding high-fitness sequences under surrogate misspecification, noise, or data scarcity

Overall, we trust our work will serve as a valuable contribution to the NeurIPS community and the field of AI for Protein Design.

---

### Decision · Program_Chairs · 2025-09-17

**Decision:**

Accept (poster)

**Comment:**

This paper introduces ProSpero, a novel active learning framework for protein design that aims to generate sequences with both high fitness and novelty by exploring beyond the local neighborhood of wild-type sequences. The method combines a pre-trained generative model with a surrogate updated via oracle feedback, using a biologically-inspired targeted masking strategy and constrained Sequential Monte Carlo sampling. The primary concern is a critical and unresolved mismatch between the paper's central claim and its validation. The paper claims to enable "robust protein design beyond wild-type neighborhoods," a significant challenge in the field. As pointed out forcefully by Reviewer HtS2, this claim cannot be substantiated by in silico benchmarks alone, as even state-of-the-art oracles like ESMFold are known to be insufficiently rugged and poor proxies for real biological function far from the training distribution.